# Phytochemical Analysis, Antioxidant Activities In Vitro and In Vivo, and Theoretical Calculation of Different Extracts of *Euphorbia fischeriana*

**DOI:** 10.3390/molecules28135172

**Published:** 2023-07-02

**Authors:** Yue Sun, Jia-Xin Feng, Zhong-Bao Wei, Hui Sun, Li Li, Jun-Yi Zhu, Guang-Qing Xia, Hao Zang

**Affiliations:** 1College of Pharmacy, Yanbian University, Yanji 133000, China; 15981309048@163.com (Y.S.); 13630304082@163.com (J.-X.F.); 2Green Medicinal Chemistry Laboratory, School of Pharmacy and Medicine, Tonghua Normal University, Tonghua 134002, China; sunhui9405@163.com (H.S.); swx0527@163.com (J.-Y.Z.); 3Institute of Scientific and Technical Information of Jilin, Changchun 130033, China; 13894531050@163.com; 4Key Laboratory of Evaluation and Application of Changbai Mountain Biological Gerplasm Resources of Jilin Province, Tonghua 134002, China

**Keywords:** *Euphorbia fischeriana*, phytochemical analysis, antioxidant activity, theoretical calculation

## Abstract

*Euphorbia fischeriana* has a long-standing history of use in traditional medicine for the treatment of tuberculosis diseases. However, the plant’s therapeutic potential extends beyond this specific ailment. The present study aimed to investigate the antioxidant properties of *Euphorbia fischeriana* and lay the groundwork for further research on its potential therapeutic applications. Phytochemical tests were performed on the plant, and 11 types of phytochemicals were identified. Ultraviolet–visible spectrophotometry was used to evaluate the active components and antioxidant properties of eight different solvent extracts, ultimately selecting acetone extract for further research. UHPLC-ESI-Q-TOF-MS identified 43 compounds in the acetone extract, and chemical calculations were used to isolate those with high content and antioxidant activity. Three stability experiments confirmed the extract’s stability, while cell viability and oral acute toxicity studies demonstrated its relatively low toxicity. In rats, the acetone extract showed significant protective effects against D-galactosamine-induced liver damage through histopathological examination and biochemical analysis. These results suggest that *Euphorbia fischeriana*’s acetone extract has potential in treating diseases related to oxidative imbalances. Therefore, this study highlights the plant’s potential therapeutic applications while providing insight into its antioxidant properties.

## 1. Introduction

*Euphorbia* is a diverse genus comprising around 2000 species, making it the largest in the spurge family and one of the largest among angiosperms. This genus has a global distribution, with a higher concentration found in Africa and Central and South America; however, 80 species are also found in China, distributed in both the northern and southern regions [1]. Genus *Euphorbia* has a long history of medicinal use, particularly in traditional medicine, whereby it has been used for treating respiratory tract infections, digestive discomfort, microbial infections, body and skin irritation, snake and scorpion bites, and body pain, among other conditions [2]. In a systematic review of *Euphorbia* plants, researchers analyzed both the chemical composition and pharmacological properties of the genus. The review revealed that *Euphorbia* plants contain a variety of valuable compounds such as diterpenoids, triterpenoids, sesquiterpenoids, steroids, and flavonoids. In addition, in the field of pharmacology, extensive research has been conducted with a focus on various activities, such as antiproliferative, multidrug resistance regulatory, cytotoxic, DNA-damaging, antiviral, PEP inhibitory, antidiarrheal, antibacterial, anti-inflammatory, antipyretic analgesic, angiotensin converting enzyme inhibitory, urease inhibitory, antitumor, and other activities [3].

*Euphorbia fischeriana*, a perennial herb and a representative plant of the *Euphorbia* genus, was first recorded in Shennong’s herbal classic. It can be found in grasslands, dry hilly slopes, rocky hillsides, and sparse pine forests, primarily on sunlit slopes at an altitude of 100–600 m. The plant is mainly distributed in Heilongjiang Province, Jilin Province, Liaoning Province, Inner Mongolia Autonomous Region, and Shandong Province in China. Additionally, it can also be found in Mongolia and Russia. The roots of the plant (Figure 1) are used traditionally for the treatment of tuberculosis diseases [4]. Moreover, *E. fischeriana* has demonstrated pharmacological effects, such as anti-tumor, antibacterial, and antiviral effects [5,6]. In recent years, its anticancer activity has gained extensive attention. Previous studies have confirmed the inhibitory effects of *E. fischeriana* on malignant melanoma, Lewis lung cancer, and ascitic liver cancer in mice [7,8,9]. The plant’s anticancer effect is primarily attributed to its diterpenoids [10], and research shows that the diterpenoids jolkinolide A and jolkinolide B, isolated from *E. fischeriana* root, have been effective in treating lung cancer [11,12]. Jolkinolide B has been found to enhance the apoptosis of human leukemia HL-60, THP-1, and U937 cells [13,14], as well as inhibit breast cancer, laryngeal cancer, and colorectal cancer [15,16,17].

Over 200 chemical constituents, including diterpenoids, triterpenoids, cyclic terpenoids, acetophenones, flavonoids, coumarins, steroids, phenolic acids, and tannins, have been identified from the roots of *E. fischeriana* [18]. Given the structure and type of compounds present in the plant, it is reasonable to infer that *E. fischeriana* possesses antioxidant properties. However, to date, only one report has been published on the antioxidant activity of the plant, demonstrating that the volatile oil extracted from its roots has a certain scavenging effect on DPPH [19]. Hence, the present study aims to investigate various solvent extracts of *E. fischeriana* and determine the most suitable solvent for extracting its antioxidant components using the content of active components and antioxidant capacity as evaluation criteria. Additionally, the study seeks to utilize UHPLC-MS technology to identify the components present in the solvent extract and calculate the antioxidant capacity of the principal components, thereby laying a foundation for the clinical application of *E. fischeriana*. Importantly, this study will significantly expand our knowledge on the potential of *E. fischeriana* as an antioxidant agent and, therefore, contribute to the development of new and improved therapeutic interventions against oxidative-stress-related diseases.

## 2. Results and Discussion

### 2.1. Qualitative Phytochemical Analysis

Table 1 shows the identification of eleven classes of phytochemicals in *Euphorbia fischeriana*. Saponins, anthraquinones, cardiac glycosides, and cyanogenic glycosides were not detected. Two interesting phenomena were observed in the conducted experiments. The first phenomenon involves the detection of flavonoids in which two results were positive and the other two were negative. This contradiction might be attributed to the low flavonoid content in plants. It may also be due to the high sensitivity and low detection limit of AlCl_3_ and lead acetate tests, but with poor sensitivity in Shinoda and alkaline reagent tests. However, this speculation needs verification through total flavonoid content determination. The second phenomenon is the positive results in all three alkaloid experiments, suggesting the presence of alkaloids in *Euphorbia fischeriana*. This new finding is novel and has not been reported in previous studies. The content of total alkaloids must still be determined.

### 2.2. Yields

Eight solvents with distinct polarities were utilized to extract *Euphorbia fischeriana* powder. Extraction yields ranged from 8.0 ± 0.1% to 33.4 ± 0.4% (*w*/*w*) (Table 2). The aqueous extract had the highest yield owing to the abundance of water-soluble constituents such as polyphenols, proteins, and carbohydrates, followed by methanol, ethanol, and acetone extracts. The dichloromethane and hexane extracts had the lowest yield. While the aqueous extract had the highest yield, it generally contains ineffective ingredients, such as pigments and pectins. Additionally, if carbohydrates are not the research object, the significant amount of carbohydrates present in the aqueous extract can interfere with the study of other components, affecting the outcomes of compound identification and biological activity. Consequently, this study did not investigate the aqueous extract, and further research on the selection of solvent extract should consider the active ingredient content and antioxidant activity outcomes.

### 2.3. Quantitative Phytochemical Analysis

#### 2.3.1. Total Carbohydrate Content (TCC)

Carbohydrates are vital components of animal and plant cells, with crucial physiological functions such as storing and supplying energy, saving protein, and preventing the production of ketone bodies [20]. Glycosides, in particular, have various pharmacological effects, as evidenced by acetophenone glycosides from *Euphorbia fischeriana* demonstrating inhibitory effects against *Mycobacterium smegmatis* [21]. This study evaluated the carbohydrate content of various solvent extracts of *Euphorbia fischeriana*, and as indicated in Table 3, the total carbohydrate content (TCC) ranged from 0.0 to 537.8 ± 19.6 mg glucose equivalents (GE)/g extract. Significant differences were observed among the groups, with the aqueous extract having the highest TCC, followed by the ethanol extract. Notably, no TCC was detected in the dichloromethane extract. This finding may be attributed to the presence of carbohydrates and glycoside components in the aqueous extract.

#### 2.3.2. Total Protein Content (TP_ro_C)

Furthermore, plant protein possesses diverse properties and nutritional value, and is readily digested and absorbed by the human body. Plant protein has been associated with a range of health benefits, including immune regulation, antioxidant, and anti-fatigue effects [22]. In the present study, the aqueous extract of *Euphorbia fischeriana* showed the highest TP_ro_C, which was 643.2 ± 12.5 mg bovine serum albumin equivalents (BSAE)/g extract (Table 3). The methanol, ethanol, and acetone extracts also showed higher TP_ro_C levels. Previous studies have shown that higher TP_ro_C values indicate the presence of more antioxidant components in the extract [23], suggesting that the ethanol and acetone extracts of *Euphorbia fischeriana* contain antioxidants.

#### 2.3.3. Total Triterpenoid Content (TT_ri_C)

Triterpenoids are important bioactive compounds that exhibit diverse biological activities, including anticancer, antiallergy, antiatherosclerosis, and antiulcer properties [24]. In the present study, the TT_ri_C was determined in eight different solvent extracts of *Euphorbia fischeriana*. As shown in Table 3, the acetone extract exhibited the highest TT_ri_C, which was 870.5 ± 49.8 mg ginsenoside Re equivalents (GRE)/g extract, followed by the ethyl acetate extract. These results of this study are consistent with a previous report indicating that *Euphorbia fischeriana* is abundant in triterpenoids [18].

#### 2.3.4. Total Phenolic Content (TP_he_C)

Plant polyphenols, such as flavonoids, are another group of bioactive substances that have excellent antioxidant capacity and display antitumor and antiviral activities. They also play a critical role in the prevention of cardiovascular disease and dementia [25] and are widely used in the cosmetic, food, and medical industries [26]. The TP_he_C in the eight solvent extracts of *Euphorbia fischeriana* was also measured in this study (Table 3). The TP_he_C values ranged from 0.0 to 38.6 ± 2.2 mg gallic acid equivalents (GAE)/g extract, with the acetone extract showing the highest TP_he_C. These results further highlight the therapeutic potential of *Euphorbia fischeriana* as a rich source of polyphenols with a wide range of biological activities.

#### 2.3.5. Total Flavonoid Content (TFC)

Flavonoids are a subclass of polyphenols that are known to prevent cell degeneration and aging, inhibit the growth of cancer cells, regulate blood pressure and cholesterol, and exhibit preventive effects against cardiovascular and cerebrovascular diseases [27]. Unfortunately, the total flavonoid content (TFC) in the different solvent extracts of *Euphorbia fischeriana* was found to be very low, ranging from 1.4 ± 0.0 to 3.2 ± 0.1 mg quercetin equivalents (QE)/g extract (Table 3). This finding can explain the contradictory results of the four qualitative experimental analyses conducted earlier, as the low content of flavonoids might have resulted in different experimental sensitivities.

#### 2.3.6. Total Tannin Content (TT_an_C), Gallotannin Content (GC), and Condensed Tannin Content (CTC)

Tannins are bioactive compounds that possess antibacterial and antiviral properties. They are also capable of removing superoxide free radicals from the body and delaying aging [28]. To evaluate the tannin content in various solvent extracts of *Euphorbia fischeriana*, the GC, CTC, and TT_an_C were determined (Table 3). The acetone extract exhibited the highest GC and CTC. The aqueous extract had the highest TT_an_C and the acetone extract had the second-highest TT_an_C. These results demonstrate the potential of *Euphorbia fischeriana* as a rich source of tannin and further support the therapeutic benefits of this plant.

#### 2.3.7. Total Alkaloid Content (TAC)

Alkaloids are a group of bioactive compounds that have a wide range of pharmacological activities, including the protection of the cardiovascular, nervous, and immune systems, and anti-cancer properties [29]. While the results of qualitative experiments on three alkaloids tested in this study were all positive, the TAC of different extracts was found to be very small, ranging from 1.6 ± 0.0 to 2.4 ± 0.0 mg berberine hydrochloride equivalents (BHE)/g extract (Table 3). This finding indicates that the detection limits of the three qualitative experiments were low and sensitive.

Overall, this study provides valuable insights into the potential of *Euphorbia fischeriana* as a source of bioactive compounds, including carbohydrates, glycosides, plant protein, phenolics, triterpenoids, tannins, and alkaloids, with various physiological and pharmacological functions. Further research is required to explore the mechanisms of action and the potential applications of these compounds in health and medical research. This could significantly contribute to our understanding of the therapeutic potential of *Euphorbia fischeriana* and the development of new treatments for various diseases.

### 2.4. Antioxidant Activity In Vitro

#### 2.4.1. DPPH and ABTS

The assessment of the free radical scavenging ability of different extracts is essential to understand the antioxidant potential of natural products. DPPH and ABTS scavenging assays are widely utilized to evaluate the free radical scavenging ability in vitro [30]. DPPH is a fat-soluble free radical while ABTS is water-soluble, and each assay targets different types of free radicals, allowing for the identification of specific antioxidants.

In our experiment, we investigated the free radical scavenging ability of various extracts of *Euphorbia fischeriana*. Our results are consistent with the existing literature. Specifically, the methanol, ethanol, and acetone extracts showed robust antioxidant capacity in DPPH assays, while the acetone and ethanol extracts also displayed strong ABTS scavenging activity (Table 4). Among the tested *Euphorbia* species, *Euphorbia ebracteolata*, *Euphorbia tirucalli,* and *Euphorbia heyneana* exhibited potent DPPH scavenging ability while *Euphorbia tirucalli* and *Euphorbia Atlantic* demonstrated strong ABTS scavenging activity [31,32,33,34].

#### 2.4.2. Hydroxyl Radicals and Superoxide Radicals

The study of hydroxyl radicals and superoxide radicals produced in the body is also crucial to evaluate the antioxidant capacity of a compound. The scavenging of these radicals showcases the ability of antioxidants to combat oxidative toxicity in cells [35].

Our findings indicated that the acetone extract of *Euphorbia fischeriana* exhibited the most robust hydroxyl radical scavenging activity, followed by the ethanol extract. Additionally, in experiments conducted with superoxide radicals, the dichloromethane extract was the most potent scavenger using curcumin as a positive control (Table 4). Previous data also indicated that *Euphorbia heyneana* possesses the ability to scavenge both hydroxyl radicals and superoxide radicals [33].

#### 2.4.3. FRAP and CUPRAC

Assessing the antioxidant capacity of samples is an important step in identifying potential therapeutic agents. One method of measuring this capacity is by evaluating the sample’s ability to reduce iron and copper ions [36]. Two commonly used methods for this evaluation are FRAP and CUPRAC assays. FRAP experiments are carried out in acidic conditions, while CUPRAC experiments are conducted in neutral conditions, closer to the physiological environment, making the latter method more reliable for assessing therapeutic potential.

In our study, we used FRAP and CUPRAC assays to evaluate the antioxidant capacity of various *Euphorbia fischeriana* extracts. The methanol, ethanol, and acetone extracts exhibited strong antioxidant capacity in FRAP assays, and these results were confirmed in CUPRAC experiments (Table 5). Our findings support existing literature, which identified *Euphorbia hirta*, *Euphorbia heterophylla*, and *Euphorbia convolvuloides* as having good antioxidant performance in FRAP and CUPRAC assays [37].

#### 2.4.4. Metal Chelating

The presence of iron and copper ions can accelerate the Fenton reaction, leading to increased oxidative stress in vivo [38]. Therefore, identifying effective metal-ion chelators is critical in the search for potential therapeutic agents.

Our research showed that the ethyl acetate extract of *Euphorbia fischeriana* exhibited the highest chelating activity toward ferrous and copper ions. The chelating activity of the other solvent extracts was lower in comparison (Table 5). Additionally, *Euphorbia neriifolia* showed strong chelating activity toward ferrous ions [39].

Overall, these results suggest the potential of *Euphorbia* species as a rich source of natural antioxidant and chelating compounds that could be developed into powerful therapeutic agents. Further research is necessary to identify the active compounds responsible for these properties and to assess their bioavailability and pharmacological effects.

#### 2.4.5. Hydrogen Peroxide (H_2_O_2_)

H_2_O_2_ is a potent oxidizing agent and a byproduct of human metabolism. In excess, it can lead to cell and tissue damage, making its direct removal essential for maintaining the body’s health [40]. Our results showed that the acetone extract of *Euphorbia fischeriana* had the highest H_2_O_2_ scavenging activity (Table 6). Another *Euphorbia* species, *Euphorbia neriifolia*, also showed significant H_2_O_2_ scavenging ability [39], suggesting that *Euphorbia* species, in general, may possess potent H_2_O_2_ scavenging capabilities.

#### 2.4.6. β-Carotene Bleaching

β-Carotene is a widely used polyene colorant that is prone to oxidation, causing it to lose its yellow color [41]. Antioxidants can slow down the bleaching rate of β-carotene by inhibiting its oxidation process. The strength of an antioxidant can be determined by the degree of decrease in β-carotene absorbance over time. In our research, we evaluated the antioxidant capacity of different extracts of *Euphorbia fischeriana* using the β-carotene bleaching assay. Our results showed that the methanol, ethanol, and acetone extracts exhibited strong antioxidant capacity (Table 6), comparable to that of synthetic antioxidants such as BHT and BHA (Figure 2). These findings suggest that *Euphorbia fischeriana* may be a potential source of natural antioxidants that have the ability to inhibit oxidation processes and preserve food quality during storage. Moreover, literature reports show that six different *Euphorbia* species also display β-carotene bleaching inhibition activity. These findings support the potential of the *Euphorbia* genus to provide natural antioxidants [42,43]. The results suggest that the *Euphorbia* species could be a promising source for developing natural antioxidants with potential for various food and medicinal applications. Further research is warranted to identify the active compounds in *Euphorbia fischeriana* and other *Euphorbia* species that contribute to the antioxidant activity.

#### 2.4.7. Singlet Oxygen

Singlet oxygen is another reactive oxygen species that can lead to cellular damage [44]. In our experiment, we found that the methanol extract of *Euphorbia fischeriana* had the best singlet oxygen scavenging activity, followed by the dichloromethane and acetone extracts (Table 6). This result suggests that *Euphorbia fischeriana* may be a valuable source of natural compounds with singlet oxygen scavenging properties.

#### 2.4.8. Hypochlorous Acid (HClO)

HClO is a potent oxidant that plays a crucial role in defending against pathogen invasion, but excessive HClO can disrupt the organism’s oxidative balance and lead to disease [45]. We found that among the *Euphorbia fischeriana* extracts, the aqueous extract had the best HClO scavenging ability, followed by the acetone extract (Table 6). This finding suggests that *Euphorbia fischeriana* extracts may contain natural compounds that could be useful in the modulation of HClO levels in the body.

#### 2.4.9. Nitric Oxide (NO)

NO is a gas that can pass freely through biofilms in the living body and participate in many biological processes, but it can also cause cellular damage by modifying protein function. Our results showed that both the aqueous and acetone extracts of *Euphorbia fischeriana* exhibited strong NO scavenging ability (Figure 3). These findings suggest that *Euphorbia fischeriana* extracts may contain natural compounds that could be useful in the regulation of NO levels in the body. The potential of *Euphorbia* species in scavenging NO is further supported by research on *Euphorbia neriifolia* and *Euphorbia tirucalli*, which also displayed robust NO scavenging ability [39,46].

Our research results demonstrate that *Euphorbia fischeriana* could emerge as a valuable source of natural compounds with potent antioxidant activity, with its acetone extract showing particularly promising results. The acetone extract displayed strong TT_ri_C, TP_he_C, GC, and CTC values. Additionally, the acetone extract also exhibited robust ABTS, hydroxyl radicals, and H_2_O_2_ scavenging abilities, together with highest FRAP and CUPRAC values. The antioxidant activity of the ethanol extract was found to be comparable to that of the acetone extract, indicating its potential for further study. However, given that terpenoids make up 70% of the more than 200 compounds isolated from Euphorbia fischeriana, they are considered the primary active components and, therefore, an important focus of research. Notably, while the ethanol extract failed to detect triterpenoids in the determination of total triterpenoids, the acetone extract exhibited the highest TTriC, suggesting that acetone is a more effective solvent for extracting terpenoids than ethanol. Based on this result, we will proceed with further research using the acetone extract. Further studies on the chemical composition of this extract will enable the identification of specific compounds responsible for its antioxidant activity. We also plan to conduct more in vivo experiments to further substantiate its antioxidant properties and assess its safety for potential use in therapeutic applications.

### 2.5. UHPLC-MS Analysis

In this study, the chemical composition of the acetone extract of *Euphorbia fischeriana* was analyzed using UHPLC–ESI–Q–TOF–MS. Molecular ions and fragment ions were matched with reference data, resulting in the identification of 39 bioactive substances (Table 7). The structures of these compounds are presented in Figure 4. The UHPLC–MS results obtained in positive-ion mode are shown in Figure 5 and the MS and MS/MS results are shown in the Appendix A.

In the UHPLC–MS results, the glycosides contained in *Euphorbia fischeriana* are vulnerable to neutral loss of different glycosyls upon collision dissociation, such as glucose (*m*/*z* 162), arabinose (*m*/*z* 132), and xylose (*m*/*z* 132). Upon analysis, peak 1 in the mass spectrum showed an ion at *m*/*z* 365.1062. The MS^2^ spectrum of this ion exhibited a fragment at *m*/*z* 162.1125 that resulted from the loss of hexose (162 Da) from the deprotonated ion at *m*/*z* 365.1062. Thus, peak 1 was identified as sucrose [47]. Peak 2 displayed [M + H]^+^ at *m*/*z* 569.2595 and produced a significant fragment ion at *m*/*z* 311.1644 ([M−C_13_H_20_O_4_−OH]^+^) and 293.1538 ([M−C_13_H_20_O_4_−OH−H_2_O]^+^, a dehydration of ortho-diols in glucose molecule), again a characteristic of phorbol-13-acetate-20-*O*-β-d-glucopyranoside [48]. Upon further analysis, peak 3 in the mass spectrum displayed an ion at *m*/*z* 477.1616. The MS^2^ spectrum of this ion showed a fragment at *m*/*z* 345.1189, which resulted from the loss of pentose (132 Da) from the deprotonated ion at *m*/*z* 477.1616. A fragment at *m*/*z* 183.0654 was obtained after the loss of hexose (162 Da) from the deprotonated ion at *m*/*z* 345.1189; thus, peak 3 was assumed to be 6-hydroxy-2-methoxy acetophenone-4-*O*-β-d-xylopyranosyl-(1→6)-β-d-glucopyranoside [21]. Peak 4 and peak 3 shared an identical MS^2^ spectrum with a fragment at *m*/*z* 345.1196 attributed to a loss of pentose (132 Da) from the deprotonated ion at *m*/*z* 477.1617. A fragment at *m*/*z* 183.0660 was also observed after hexose (162 Da) loss from the deprotonated ion at *m*/*z* 345.1196. Finally, a water molecule loss led to the fragment *m*/*z* 165.0551, tentatively identifying peak 4 as 6-hydroxy-2-methoxy acetophenone-4-*O*-α-L-arabinofuranosyl-(1→6)-β-d-glucopyranoside [21]. Peak 5 also exhibited loss of a pentose and a CH_3_ to yield a fragment *m*/*z* 345.1188, and then lost a hexose to obtain a fragment *m*/*z* 183.0654, characteristic of 2,4-dihydroxy-6-methoxy-3-methylacetophenone-4-*O*-α-L-arabinofuranosyl-(1→6)-β-d-glucopyranoside [21]. Peak 6 had an *m*/*z* of 193.0497 with an MS^2^ ion at *m*/*z* 121.0289, resulting from the RDA cleavage reaction and the loss of −CO and −OH. Peak 6 was tentatively identified as either scopoletin or isoscopoletin [49,50]. Peak 7 had [M + H]^+^ at *m*/*z* 183.0654 and yielded a significant fragment ion at *m*/*z* 165.0550 ([M−OH]^+^) and 153.0184 ([M−OCH_2_+H]^+^), which was characteristic of 2,4-dihydroxy-6-methoxyacetyl benzene [51]. Peak 8 had an *m*/*z* of 533.2365 with MS^2^ ions at *m*/*z* 295.1694 ([M−hexose−C_4_H_6_+H]^+^). Initially, pentose fragment loss occurred, and C_4_H_6_ was then lost by cyclohexanone cleavage. Peak 8 was tentatively characterized as 4β,9α,20-trihydroxy-13,15-secotiglia-1,6-diene-3,13-dione 20-*O*-β-d-glucopyranoside [52]. Peak 9 had an *m*/*z* of 329.1756 with MS^2^ ions at *m*/*z* 313.1725 ([M−CH_3_]+, this fragment is produced by ethylene oxide cracking) and was tentatively identified as euphonoid B based on the literature [53]. Peak 10 had an *m*/*z* of 197.0813. Major fragments included *m*/*z* 179.0705 ([M−OH]^+^), 165.0542 ([M−OH−CH_3_+H]^+^), and 153.0189 ([M−COCH_3_]^+^), assigned as 2,4-dihydroxy-6-methoxy-3-methyl acetophenone [54].

Peak 11 appeared at an *m*/*z* of 223.0601 with MS^2^ ions at *m*/*z* 208.0361 ([M−CH_3_+H]^+^) and 179.0701 ([M−CH_3_−CO]^+^, CO fragment produced by cleavage of hexadiene lactone), identified as fraxidin [55]. Peak 12 was observed at *m*/*z* 359.1340 and characterized as 2,4-dihydroxy-6-methoxyl-3-methyl-acetophenone-4-*O*-β-d-glucopyranoside. Its MS^2^ ions at *m*/*z* 315.0715, 197.0809, and 179.0703 implied the loss of −COCH_3_, −hexose+H, and −hexose−OH, respectively [56]. Peak 13 had an *m*/*z* of 569.2597 with a fragment ion at *m*/*z* 511.1779 after the elimination of −OCOCH_3_+H. Therefore, peak 13 was identified as phorbol-13-acetate-20-*O*-β-d-glucopyranoside [48]. Peak 14 had an *m*/*z* of 553.2663 with a major fragment at *m*/*z* 277.1587 ([M−hexose−OCOCH_3_−C_3_H_3_O]^+^, after losing hexose and −OCOCH_3_; subsequently, α,β-cyclopentenone cleavage lost fragment C_3_H_3_O), assigned as prostratin 20-*O*-β-d-glucopyranoside [48]. Peak 15 at *m*/*z* 501.2465 was proposed as 19-*O*-β-d-glucopyranosyl-ent-atis-16-ene-3,14-dione, and the main MS^2^ ions at *m*/*z* 317.2113 and 299.2002 corresponded to the loss of −hexose+H and −hexose−OH [57]. Peak 16, a [M + H]^+^ ion at *m*/*z* 705.2777, was suggested to be either 20-*O*-(4′-galloyl)-β-d-glucopyranoside or 20-*O*-(3′-galloyl)-β-d-glucopyranoside. Its MS^2^ ion at *m*/*z* 1426.5730 ([2M + H_2_O]^+^) and 153.0185 (galloyl group) corresponds to the literature [58]. Peak 17 at *m*/*z* 365.1961 yielded fragment ions at *m*/*z* 347.1861 and 329.1754, indicating losses of −OH and −OH−H_2_O, respectively. Therefore, peak 17 was assigned to langduin B [59]. Peak 18 was identified as 3S,16S,17-trihydroxy-2-one-ent-kaurane at *m*/*z* 337.2381, with characteristic fragment ions at *m*/*z* 359.2201 ([M + Na]^+^), 319.2276 ([M−OH]^+^), and 277.1590 ([M−C_2_H_4_O_2_+H]^+^) [60]. Peak 19 with *m*/*z* 373.2025 and MS^2^ ions at *m*/*z* 295.1702 ([M−C_3_H_3_O]^+^, where C_3_H_3_O was a fragment generated by the cleavage of pentylene lactone) was tentatively identified as morinda officinalis B/yuexiandajisu E/yuexiandajisu D based on the literature [61,62,63]. Finally, peak 20 had an ion at *m*/*z* 364.2483 and its MS2 spectrum exhibited fragments at *m*/*z* 148.0756, 131.0492, and 105.0697.

Peak 21 was identified as fischerianoid A with an *m*/*z* of 335.2217. The MS^2^ spectrum showed a connection of *m*/*z* 317.2116 with the loss of a hydroxyl group [64]. Peak 22 had an *m*/*z* of 317.2112 with MS^2^ ions at *m*/*z* 339.1936 ([M + Na]^+^), 289.2167 ([M−CO]^+^, cyclohexanone cracking), and 261.1852 ([M−2CO]^+^, another cyclohexanone cracking), tentatively characterized as ent-(13S)-13-hydroxyatis-16-ene-3,14-dione [53]. Peak 23 displayed a precursor ion [M + Na]^+^ at *m*/*z* 343.2254, producing a major fragment ion at *m*/*z* 321.2433 ([M + H]^+^) and 303.2327 ([M−OH]^+^) corresponding to ent-kaurane-3-oxo-16α,17-diol or ent-kaurane-3-oxo-16β,17-diol [65]. Peak 24 had a precursor ion [M + H]^+^ at *m*/*z* 379.2120, producing a major fragment ion at *m*/*z* 361.2024 ([M−OH]^+^), 333.2067 ([M−OH−CO]^+^, cleavage of pentylene lactone to produce M−CO ion), and 319.2278 ([M−OH−CO−CH_3_+H]^+^). It was tentatively characterized as fischeriabietane E [66]. Peak 25 had an *m*/*z* of 319.2265 and was identified as araucarone based on the characteristic fragment ion at *m*/*z* 181.1018 ([M−C_9_H_14_O+H]^+^) [53]. Peak 26 had an *m*/*z* of 349.2011 with MS^2^ ions at *m*/*z* 367.1529 ([M + H_2_O + H]^+^), 331.1913 ([M−OH]^+^), and 313.1809 ([M−OH−H_2_O]^+^, obtained by dehydration of ortho-diols), assigned as 7β,11β,12β-trihydroxy-ent abieta-8(14),13(15)-dien-16,12-olide [67]. Peak 27 had an *m*/*z* of 503.2412 and was identified as jolkinol A based on the characteristic fragment ions at *m*/*z* 481.2593 ([M + H]^+^), 463.2489 ([M−OH]^+^), and 131.0492 (cinnamoyl group) [63]. Peak 28 had a precursor ion [M + Na]^+^ at *m*/*z* 369.1675 and was identified as 17-hydroxyjolkinolide B [68]. Peak 29 had [M + H]^+^ at *m*/*z* 333.2070 and yielded a significant fragment ion at *m*/*z* 315.1961 ([M−OH]^+^) and 287.2374 ([M−OH−CO]^+^), which was characteristic of 13β-hydroxy-7-oxobiet-8(14)-en-19,6β-olide [69]. Peak 30 had an *m*/*z* of 487.2457 and was identified as jolkinol B, with fragment ions at *m*/*z* 447.2540 and 419.2247 corresponding to the loss of −OH and −OH−CO [70].

Peak 31 had an *m*/*z* of 505.1839 and MS^2^ ions at *m*/*z* 483.2018 ([M + H]^+^) and 331.1813 ([M−C_8_H_7_O_3_]^+^), assigned as fischeriana A [71]. Peak 32 had an [M + Na]^+^ ion at *m*/*z* 353.1726, producing a major fragment ion at *m*/*z* 331.1912 ([M + H]^+^) and 303.2324 ([M−CO+H]^+^, cleavage of pentylene lactone to produce M−CO ion), which was characteristic of jolkinolide B [68]. Peak 33 had an [M + H]^+^ ion at *m*/*z* 317.2117, producing major fragment ions at *m*/*z* 339.1935 ([M + Na]^+^) and 299.2013 ([M−OH]^+^), which was characteristic of ent-13-hydroxyatis-16-ene-3,14-dione [62]. Peak 34 was at *m*/*z* 301.2162 and characterized as ent-atis-16(17)-ene-3,14-dione or ent-atis-16-ene-3,14-dione, where its MS^2^ ions at *m*/*z* 285.2216 and 271.2425 implied the loss of −CH_3_ and −2CH_3_+H, respectively [72]. Peak 35 appeared at *m*/*z* 509.2170 with fragment ions at *m*/*z* 491.2072 ([M−OH]^+^) and 463.2122 ([M−OH−CO]^+^, cleavage of hexene lactone to produce M−CO ion), assigned as fischernolide D [73]. The mass spectrum of peak 36 showed an ion at *m*/*z* 331.1913, and its MS^2^ spectrum exhibited a fragment at *m*/*z* 317.2119, 299.2029, and 277.2163. Peak 37 had an *m*/*z* of 303.2318 and gave fragment ions at *m*/*z* 285.2221 and 257.2264, which were correlated with the loss of M−OH and M−OH−CO (α,β-cyclohexenone cleavage generates M−CO ion), respectively. Therefore, peak 37 was presumed to be ent-10α-hydroxy-rosa-1,15-dien-3-one [74]. Peak 38 was proposed as euphonoid A by matching fragment ions (*m*/*z* 298.1891 [M−OH−OCH_3_]^+^ and *m*/*z* 269.1904 [M−OH−OCH_3_−CO]^+^, the cleavage of pentylene lactone generates M−CO ion) with reference [53]. Finally, peak 39 had an *m*/*z* of 525.2126 and gave fragment ions at *m*/*z* 481.2230, which was correlated with the loss of −COCH_3_. Thus, peak 39 was presumed to be landuin D [75].

### 2.6. Molecular Electrostatic Potential (MEP) Surface Map

In this study, ten large peaks were selected based on their peak area from large to small, and their corresponding compounds were numbered as compounds **1**–**15**. A detailed overview of the compounds and their corresponding numbers can be found in Table 8 and Figure 6.

Figure 7 presents MEP surface maps of these fifteen compounds. The MEP surface maps are an excellent visual tool for understanding the relative polarity of each molecule. The MEP is a measure of the electrostatic potential energy at the surface of a molecule. Different colors are used in Figure 7 to depict the various MEP values, with blue representing the regions of the most positive electrostatic potential and red indicating the regions of the most negative electrostatic potential.

The principle of an antioxidant is to reduce the concentration of oxygen, and the activated site for antioxidant activity is expected to be the region of positive electrostatic potential. Darker areas of blue signify a more potent antioxidant activity since they exhibit higher positive electrostatic potential. As illustrated in Figure 7, compounds **9**, **10**, **14**, and **15** possessed a more positive electrostatic potential than the other compounds, suggesting that they might be better antioxidants.

### 2.7. Frontier Molecular Orbital

The frontier molecular orbital theory is a powerful tool in predicting the chemical reactivity of a given system. Investigations into the energy levels of frontier molecular orbitals offer valuable insights into the reactivity of bioactive molecules [76]. This theory is based on studying the highest occupied molecular orbital (HOMO) and the lowest unoccupied molecular orbital (LUMO) of molecules. The energies associated with the Highest Occupied Molecular Orbital (HOMO) and the Lowest Unoccupied Molecular Orbital (LUMO) play a crucial role in determining the reactivity of molecules, as they typically participate in chemical reactions [77]. The HOMO of a molecule determines its antioxidant capacity, while the strength of its oxidation capacity depends on the energy of the LUMO. Therefore, the lower the energy of the LUMO, the stronger the oxidation capacity, and the higher the energy of the HOMO, the stronger the antioxidant capacity of the molecule. The energy difference between the HOMO and LUMO is referred to as the “band gap.” To determine the band gap—the energy difference between the HOMO and the LUMO—an optimized geometry was utilized, and calculations were performed using density functional theory [78]. A smaller band gap indicates that the molecule is easier to excite.

This study aimed to explore the antioxidant activities of the fifteen compounds corresponding to the top ten peaks. To compare their antioxidant capacity, we selected two positive controls, trolox and gallic acid, which were previously used in the experiments. According to theoretical calculations, all fifteen compounds had a HOMO distribution ranging from −5.98 eV to −7.93 eV, indicating that they exhibit a certain antioxidant capacity. Among them, compounds **9** (−6.07 eV) and **14** (−5.98 eV) displayed superior HOMO performance compared to gallic acid (−6.12 eV) but were weaker than trolox (−5.42 eV). The findings of this study indicate that the presence of phenolic hydroxyl groups and glycoside groups in the compound structure plays a crucial role in increasing the energy levels of HOMO orbitals, thereby enhancing the antioxidant potential of these compounds. The band gap, which determines the chemical reactivity, is closely associated with the ease of excitation of the investigated molecules. Analysis of Figure 8 reveals that compounds **1**, **5**, **8**, **9**, **14**, and **15** exhibit higher antioxidative potential, as their band gap values (4.76 eV, 4.27 eV, 4.47 eV, 4.59 eV, 4.29 eV, and 4.59 eV, respectively) are lower than those of trolox (5.02 eV) and gallic acid (4.80 eV).

In conclusion, based on these calculations, compounds **9** and **14** demonstrate strong antioxidant capacity and are more likely to engage in chemical reactions, suggesting that they may serve as primary antioxidant components in *Euphorbia fischeriana*.

### 2.8. Stability Studies of Acetone Extract

Our study explored the stability of the acetone extract of *Euphorbia fischeriana* and its antioxidant properties through a series of experiments. Figure 9, Figure 10 and Figure 11 depict the outcomes of these experiments. We discovered that the extract’s TP_he_C value and ABTS scavenging activity were generally stable when subjected to changes in pH. The highest value for TP_he_C of the extract was noted at pH 7, and it decreased slightly with an increase or decrease in pH. In comparison, the ABTS scavenging activity showed a gradual decrease with increasing pH. The stronger the alkalinity, the greater was the impact on the acidic system of the ABTS experiment, which likely accounted for the observed decrease. Regarding heating time, we observed a slight reduction in the TP_he_C value and ABTS scavenging activity of the extract. However, the impact was minimal. Concerning stability experiments using an in vitro simulation of the human digestive system, the TP_he_C value of the acetone extract decreased gradually with time. We concluded that gastric acid, pepsin, trypsin, pancreatin, and bile may all have affected the extract, leading to the gradual decrease in the TP_he_C value. The ABTS scavenging activity of the extract followed a similar trend as the TP_he_C. Nevertheless, our stability studies revealed that the antioxidant components of the acetone extract of *Euphorbia fischeriana* were stable, which is promising for maintaining its efficacy under various physiological conditions.

### 2.9. Cell Viability

Our experiment examined the effects of the acetone extract of *Euphorbia fischeriana* on the cellular morphology of TM_3_ mouse Leydig cells after 24 and 48 h of treatment, as shown in Figure 12. The results indicated that at low doses (25 and 50 μg/mL), the acetone extract did not exhibit any cytotoxicity. In fact, it even promoted cell proliferation to a certain extent with increasing incubation time. However, at moderate doses (100 μg/mL), the extract induced some cytotoxicity, but this did not increase with the extension of incubation time. On the other hand, at high doses (200 μg/mL), the extract demonstrated clear cytotoxicity, and its toxic effects increased significantly with the extension of incubation time (Table 9). These results emphasize the importance of dosage when using *Euphorbia fischeriana* as a therapeutic agent. Our findings suggest that *Euphorbia fischeriana* extract has the potential to be developed into an effective drug for certain health conditions. However, it is necessary to pay careful attention to the dose administered to avoid possible toxic effects. It is worth noting that different *Euphorbia* species have varying degrees of cytotoxicity that can affect different types of cells. For instance, *Euphorbia vajraveleu* has been found to be non-toxic to normal cells, H9C2, and has negligible toxic effects on cervical cancer cells, Hela [79]. Conversely, *Euphorbia lathyris*, another species of this genus, has been found to show strong cytotoxicity against a variety of cancer cells [80]. These differences in cytotoxicity among *Euphorbia* species may be attributed to the varying chemical compositions and concentrations of their extracts.

### 2.10. Oral Acute Toxicity Study

In our study, we administered a single dose of 2000 mg/kg of *Euphorbia fischeriana* acetone extract orally to a group of mice, and noted that none of the 20 tested mice died within 24 h. This finding suggests that the toxicity associated with acetone extract is relatively low. Similarly, a study on *Euphorbia fusiformis* found that mice tolerated up to 5000 mg/kg bw without any incidence of mortality. The single dose LD_50_ was determined to be 10,000 mg/kg bw in this study [81]. Another study evaluated the safety of *Euphorbia hirta* extract in rats and found that a single dose of 5000 mg/kg did not induce any significant side effects or mortalities during the 14-day observation period [82]. Taken together, these studies provide compelling evidence indicating that *Euphorbia* plants are relatively safe for use in rats and mice. However, it should be noted that these findings do not necessarily imply that *Euphorbia* plants are safe for human consumption. To date, only a limited number of clinical studies have been conducted on *Euphorbia* extracts. More research, including preclinical and clinical studies, is needed to evaluate their safety and efficacy in humans.

### 2.11. Hepatoprotective Activity

The liver is a critical metabolic organ in vertebrates that plays a vital role in processing nutrients, storing energy, and eliminating waste substances. When the liver is damaged, it can lead to a decrease in the fluidity of the liver cell membrane, which causes an increase in cell permeability. As a result, enzymes such as aspartate aminotransferase, alanine aminotransferase, and γ-glutamyl transpeptidase are released into the bloodstream. Liver damage can also impair the binding and excretion of bilirubin in bile, which increases the concentration of total bilirubin in the blood [83].

To examine the effectiveness of *Euphorbia fischeriana* acetone extract in managing liver damage, we conducted experiments on rats. Forty rats were randomly divided into five groups (n = 8). Each group received a different treatment orally. The control group (group I) was given 0.5% carboxymethylcellulose sodium. The negative control group (group II) was administered D-galactosamine and 0.5% carboxymethylcellulose sodium. The high-dose group (group III) received *Euphorbia fischeriana* acetone extract at 300 mg/kg BW. The low-dose group (group IV) received *Euphorbia fischeriana* acetone extract at 150 mg/kg BW. Finally, the comparison group (group V) was given silymarin at 100 mg/kg BW.

We fed the rats with *Euphorbia fischeriana* acetone extract every day for seven days by gavage, followed by D-galactosamine injection. The results showed that pretreatment with the acetone extract (high and low dose groups) significantly reduced the viscera index in the rats compared to group II, indicating an improvement in liver condition (*p* < 0.001, Figure 13). Moreover, the high-dose group had a significantly better effect on liver function than silymarin (*p* < 0.001, Figure 13).

Our results revealed that compared with group II, the activity levels of liver enzymes including aspartate aminotransferase, alanine aminotransferase, and γ-glutamyl transpeptidase were significantly reduced in the groups treated with *Euphorbia fischeriana* acetone extract (*p* < 0.001, shown in Figure 14). In particular, compared to group II, the reduced activity of alanine aminotransferase was 63.57% and 68.17% in groups III and IV, respectively, while aspartate aminotransferase was reduced by 61.54% and 62.72%, respectively. These findings suggest that the low-dose acetone extract of *Euphorbia fischeriana* has a potent curative effect on liver damage caused by D-galactosamine.

The reduction in the activity of liver enzymes observed in our study is indicative of the amelioration of liver damage. Increased activity levels of these enzymes in the liver are an indication of liver inflammation and injury. Therefore, the significant reduction in the activity of aspartate aminotransferase, alanine aminotransferase, and γ-glutamyl transpeptidase in the treatment groups supports the use of *Euphorbia fischeriana* acetone extract as a potential therapeutic agent for liver-related diseases and disorders.

Moreover, compared with group II, the albumin level was significantly reduced in the rats, while treatment with both silymarin and the acetone extract significantly increased the albumin level after 24 h of modeling (*p* < 0.001). The low-dose group was better than the high-dose group in improving albumin levels. Additionally, D-galactosamine injection increased the concentration of total bilirubin in the blood of rats in group II, while the concentration of total bilirubin in groups III–V decreased significantly compared to group II, with the low-dose group performing better than the high-dose group and being comparable to silymarin. Furthermore, D-galactosamine-induced liver injury increased the production of reactive oxygen species and reduced the efficacy of antioxidants in vivo. This led to an increase in the level of malondialdehyde and a decrease in the level of glutathione in the liver of rats. Compared with group II, the effect of the acetone extract on enhancing glutathione levels was not ideal, but it significantly reduced malondialdehyde levels (*p* < 0.001, shown in Figure 15). The low-dose group performed better than the high-dose group and was comparable to silymarin in reducing malondialdehyde levels.

In summary, our findings indicate that the low-dose acetone extract of *Euphorbia fischeriana* has a good liver-protective effect through antioxidant activity. The overall protective effect of this extract is comparable to that of silymarin, suggesting its potential as a therapeutic agent for liver-related disorders and diseases.

The results of the histopathological examination are presented in Figure 15. In the control group, hepatocytes were arranged in a normal pattern, and no inflammatory cell infiltration was observed around the portal area (Figure 16A). However, in group II, the liver’s histological structure was disordered, the hepatic cord was absent, single-cell necrosis was visible (no-tailed arrow), and a large number of inflammatory cells were present (long-tailed arrow) (Figure 16B). The magnified view of group II at 400× further revealed the presence of increased inflammatory cell infiltration and necrosis (Figure 16C). Remarkably, the high-dose group showed significant improvement in hepatocyte injury with fewer inflammatory and necrotic cells (Figure 16D). Similarly, the low-dose group demonstrated better results compared to the high-dose group, with fewer inflammatory and necrotic cells (Figure 16E). It was interesting to note that the positive group had the best curative effect, which was evident by the significant reduction in inflammatory and necrotic cells and the relatively complete morphology of hepatocytes (Figure 16F).

It is essential to understand the significance of the findings observed in this study. The histopathological examination reveals the extent of damage to the liver caused by D-galactosamine. The presence of inflammatory cells, necrotic cells, and a disordered histological structure in group II demonstrates the severity of liver damage induced by D-galactosamine. The reduction in these parameters in the positive, high-dose, and low-dose groups indicates the hepatoprotective effect of the Euphorbia fischeriana acetone extract. These findings support the potential use of Euphorbia fischeriana extract in the treatment of liver disorders.

Recent research has shown that different species of *Euphorbia* plants have significant hepatoprotective effects. For instance, *Euphorbia fusiformis* has been found to decrease elevated biochemical parameters to a level comparable to that of the control group [78]. Another study conducted on *Euphorbia antiquorum* demonstrated that it significantly increased the level of reduced glutathione in tissues by reducing the activities of serum enzymes, bilirubin, triglycerides, and lipid peroxidation in rats. Furthermore, its hepatoprotective and antioxidant activities were comparable to those of silymarin [84].

These observations suggest that *Euphorbia* plants could be potential sources of natural liver protective agents and antioxidants. The hepatoprotective properties of these plants are likely due to the antioxidant principle and potential of the ingredients they contain. Our study found that *Euphorbia fischeriana* acetone extract has a beneficial effect on liver health, which suggests its potential therapeutic applications in managing liver-related disorders. However, further research is necessary to determine the bioactive components responsible for these beneficial effects, as well as the optimal dosage and course of treatment. It is essential to identify the optimal dosage and timing to ensure the extract’s maximum therapeutic benefits while minimizing any potential side-effects. Clinical studies are also necessary to validate the results of our study and provide more precise recommendations for using the extract in human patients. In conclusion, our results highlight the potential of *Euphorbia* plants as a natural source of liver-protective agents and antioxidants. Our findings indicate that *Euphorbia fischeriana* acetone extract has a beneficial effect on liver health and supports further exploration of its therapeutic potential in managing liver-related diseases and disorders. We hope this study will inspire more research and clinical studies on *Euphorbia* plants as natural therapeutic agents for liver-related ailments.

## 3. Materials and Methods

### 3.1. Reagents and Chemicals

3-(4,5-dimethylthiazol-2-yl)-2,5-diphenyl-2H-tetrazolium bromide (MTT), S-butyrylthiocholine chloride, and *p*-nitroblue tetrazolium chloride (NBT) were purchased from Sigma-Aldrich. Curcumin, salicylic acid, *L*-ascorbic acid, 2,4,6-tri(2-pyridyl)-s-triazine (TPTZ), ammonium acetate (NH_4_Ac), cupric sulphate, ferrous sulfate heptahydrate (FeSO_4_·7H_2_O), copper sulphate (CuSO_4_), taurine, 4-aminoantipyrine, lipoic acid, ferulic acid, sulfanilamide, cupric chloride dihydrate (CuCl_2_·2H_2_O), phosphoric acid (H_3_PO_4_), ninhydrin hydrate, quercetin, naphthylethylenediamine dihydrochloride, D-(+)-glucose, butylated hydroxytoluene (BHT), 2,9-dimethyl-1,10-phenanthroline (Neocuproine, Nc), *α*-naphthol, iodine, tertiary butylhydroquinone (TBHQ), 3,5-dinitrosalicylic acid (DNS), gelatin, potassium iodide (KI), ferric chloride (FeCl_3_), 4-nitroaniline, sodium nitrite, antimony trichloride, calcium hydroxide (Ca(OH)_2_), ABTS, copper sulfate pentahydrate (CuSO_4_·5H_2_O), phosphomolybdic acid hydrate, hydroxylamine hydrochloride, potassium hydroxide, vanillin, 3,5-dinitrobenzoic acid, phenol, dipotassium hydrogen phosphate, potassium dihydrogen phosphate, sodium dihydrogen phosphate, dibasic sodium phosphate, sodium nitroprusside dehydrate, sodium hypochlorite (NaClO) (10% active chloride), tannic acid, potassium persulfate, potassium chloride (KCl), sodium acetate, gallic acid, sodium molybdate, arbutin, L-tyrosine, urea, ginsenoside Re, phloroglucinol, potassium iodate, oleanolic acid were purchased from Energy Chemical. Benedict’s Reagent was purchased from Adamas. DPPH was purchased from Alfa Aesar. *β*-carotene, bromocresol green, trolox, pyrocatechol violet, sudan III, and sudan IV were purchased from TCI. Folin & Ciocalteu’s phenol reagent (FC reagent), aluminum chloride hexahydrate (AlCl_3_·6H_2_O), linoleic acid, 3-(2-pyridyl)-5,6-diphenyl-1,2,4-triazine-4′,4″-disulfonic acid sodium salt (Ferrozine), ferrous chloride tetrahydrate (FeCl_2_·4H_2_O), sodium potassium tartrate tetrahydrate (Rochelle salt), ethylenediaminetetraacetic acid disodium salt dihydrate (EDTANa_2_·2H_2_O), potassium ferricyanide (K_3_[Fe(CN)_6_]), Lead(II) acetate trihydrate, tungstosilicic acid hydrate, bismuth subnitrate, mercury(II) chloride (HgCl_2_), pepsin (32 U/mg), pancreatin, bovine bile extract, magnesium acetate, sodium thiosulfate standard solution (0.1 M), potassium hydroxide standard solution (0.1 M), phenolphthalein, tween 40, and 1,3-dinitrobenzene were purchased from Xiya Reagent. Concentrated sulfuric acid (H_2_SO_4_), phenol, sodium carbonate (Na_2_CO_3_), methanol, ethanol, acetone, ethyl acetate, dichloromethane, hexane, dimethyl sulfoxide (DMSO), petroleum ether (60–90 °C), sodium hydroxide (NaOH), concentrated hydrochloric acid (HCl), sodium chloride (NaCl), magnesium powder, acetic acid, ammonium hydroxide (NH_3_·H_2_O), acetic anhydride, 30% hydrogen peroxide (H_2_O_2_), formaldehyde, and 3% bromine water were purchased from Sinopharm. All reagents and solvents used were analytical grade. Litmus paper blue was purchased from Tianjin Jinda Chemical Reagent Co., Ltd. BCA kit was purchased from Beyotime. High-glucose Dulbecco’s modified Eagle’s medium (HG-DMEM) and 100× penicillin-streptomycin solution were purchased from Hyclone. Fetal bovine serum (FBS) was purchased from Bioind. TM_3_ mouse leydig cells were purchased from Cell Bank of the Chinese Academy of Sciences. Trypsin (2500 U/mg) was purchased from Aladdin (Bay City, MI, USA).

### 3.2. Materials

*Euphorbia fischeriana* was gathered (voucher specimen number: 2021-05-03-001) in Baicheng (latitude N 45°45′48.92″, longitude E 121°39′33.04″, altitude 502.0 m, Jilin Province, China) in May 2021. Professor Junlin Yu identified the specimens. The voucher specimen is stored in the Herbarium of Tonghua Normal University.

### 3.3. Qualitative Phytochemical Analysis

Qualitative phytochemical analysis was performed on 15 types of chemical components, following a previously established method [23]. The detailed experimental procedure is described in the Appendix A.

### 3.4. Quantitative Phytochemical Analysis

Quantitative phytochemical analysis was conducted to determine the concentration of various compounds such as TCC, TP_ro_C, TT_ri_C, TP_he_C, TFC, TT_an_C, CTC, GC, and TAC, using methods previously described in reference [23]. The detailed experimental procedure is described in the Appendix A.

### 3.5. Antioxidant Activity Assays

Antioxidant activity assays were performed using a range of different methods, including DPPH, ABTS, hydroxyl radicals, superoxide radicals, FRAP, CUPRAC, metal chelating, H_2_O_2_, HClO, β-carotene bleaching, and NO. These assays were conducted following previously established protocols [23]. The detailed experimental procedure is described in the Appendix A.

### 3.6. UHPLC-MS

Acetone extract of *Euphorbia fischeriana* was analyzed using UHPLC (Agilent 1290 system) with Q-TOF-MS (Agilent 6545 system). A ZORBAX SB-C_18_ column (150 × 3.0 mm, 1.8 µm; Agilent) was used. The column temperature was set to 40 °C. The mobile phase was a mixture of 0.1% formic acid in water (solvent A) and a mixture of 0.1% formic acid in acetonitrile (solvent B) at a flow rate of 0.4 mL/min. Linear gradient elution was applied (0–1 min, 95% A; 1–30 min, 95–70% A; 30–50 min, 70–30% A; 50–56 min, 30–1% A; 56–60 min, 1% A). The extract was diluted to 1 mg/mL with methanol and filtered using a 0.22 µm membrane before use. The sample injection volume was 5 µL. The Q-TOF-MS (Agilent) was operated in positive-ion mode with scan range *m*/*z* 100–1700. Data were recorded and analyzed with Qualitative Analysis software (version B. 07.00, Agilent).

### 3.7. Computational Methods

All calculations were performed using the Gaussian 09 program package at the B3LYP-D3/6-311G (d, p) level [85,86,87,88,89]. Following the optimization of the molecular structures, frequency calculations were performed to ensure that the optimized structures corresponded to minimum energy points with no virtual frequencies. Furthermore, the MEP surface of compounds **1**–**15** were analyzed using MULTIWFN software and the VMD program, while the frontier molecular orbitals (HOMO and LUMO) were analyzed using GaussView [90,91,92].

### 3.8. Cell-Viability Assay

To evaluate the cell viability, the TM_3_ mouse cell line was used in the MTT assay, which was carried out as described in reference [23].

### 3.9. Oral Acute Toxicity Study

An oral acute toxicity study was conducted according to previously established methods [93].

### 3.10. Hepatoprotective Experiments

The hepatoprotective experiments performed included animal selection, experimental protocols, histopathological examination, and biochemical analyses, following the procedures outlined in reference [93].

### 3.11. Statistical Analysis

Statistical analysis was performed to assess the significance of the data. The data were presented as means with the standard error of the mean. One-way analysis of variance with post hoc least significant difference tests was used to test for significant correlations between groups. Pearson’s correlation analysis was used to investigate the relationship between antioxidant activity and total active constituents. *p*-values of 0.05, 0.01, and 0.001 were considered significant, highly significant, and very highly significant, respectively.

## 4. Conclusions

*Euphorbia fischeriana* is a well-known medicinal plant that has been used in traditional medicine for many years. Despite the numerous reports on the components of this plant, its antioxidant activity in vivo and in vitro and the identification of compounds that have antioxidant activities are still unknown. Therefore, this study aimed to fill this knowledge gap by investigating the antioxidant properties of *Euphorbia fischeriana*. The results of phytochemical analysis showed that *Euphorbia fischeriana* contains 11 types of phytochemicals. The contents of active components and antioxidant properties were evaluated in eight different solvent extracts of *Euphorbia fischeriana* using ultraviolet–visible spectrophotometry. Among these, the acetone extract exhibited the highest contents of active components and antioxidant activity and was selected as the object of further study. Further analysis of the acetone extract led to the identification of 43 specific compounds. To determine which compounds have antioxidant activities, the top ten peaks were selected for theoretical calculations of antioxidant capacity. The results confirmed the presence of antioxidant components in *Euphorbia fischeriana* acetone extract and helped clarify the antioxidant mechanism of this plant. To evaluate the effectiveness of *Euphorbia fischeriana* acetone extract as a potential antioxidant agent, its stability and antioxidant capacity were evaluated during heating, at different pH values, and after in vitro digestion. The results showed that the acetone extract exhibited excellent stability and antioxidant capacity even in adverse conditions, indicating its potential therapeutic applications. Finally, in vivo antioxidant experiments were conducted, and the results showed that the low-dose acetone extract displayed a significantly better protective effect on liver injury in rats.

In conclusion, our findings demonstrate that *Euphorbia fischeriana* acetone extract contains compounds with potent antioxidant properties. The identification of these active components provides a foundation for further exploration of this plant for its therapeutic potential in treating diseases related to oxidative stress. However, given the potential cytotoxicity of *Euphorbia* species, more in-depth research is needed to ensure their safe and effective use. We hope that our study will inspire further research on the antioxidant properties of *Euphorbia fischeriana* and contribute to the development of novel natural antioxidants for human health and wellbeing.

## Figures and Tables

**Figure 1 molecules-28-05172-f001:**
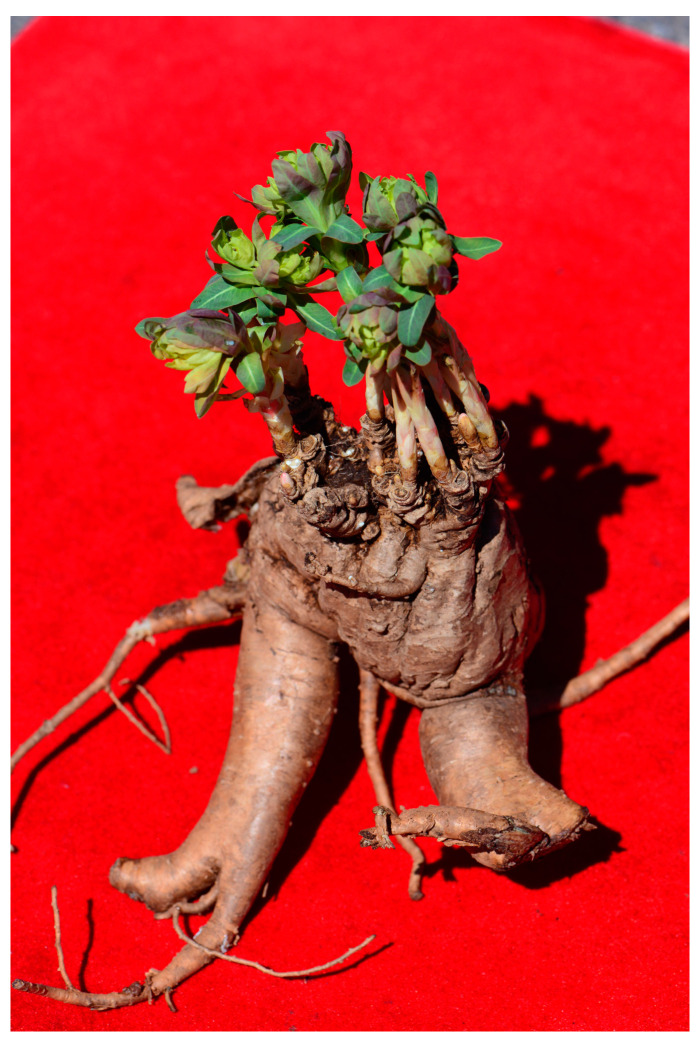
Morphology of *Euphorbia fischeriana*.

**Figure 2 molecules-28-05172-f002:**
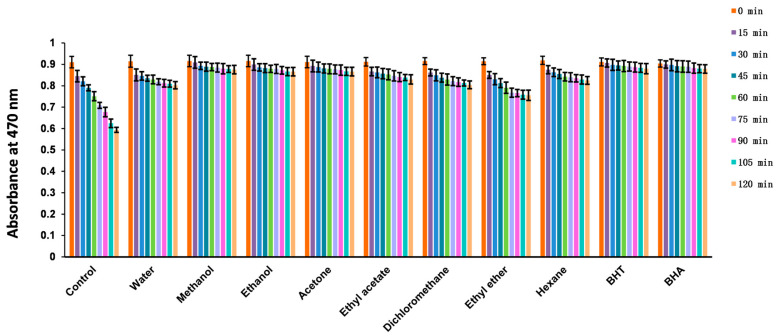
Changes in the absorbance of β-carotene in the presence of solvent extracts of *Euphorbia fischeriana*. BHT: Butylated hydroxytoluene; BHA: Butyl hydroxyanisole.

**Figure 3 molecules-28-05172-f003:**
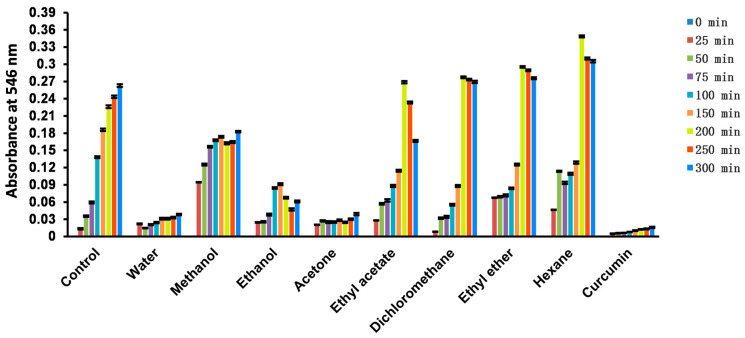
Changes in the absorbance of solvent extracts of *Euphorbia fischeriana* over time measured using the nitric oxide scavenging method.

**Figure 4 molecules-28-05172-f004:**
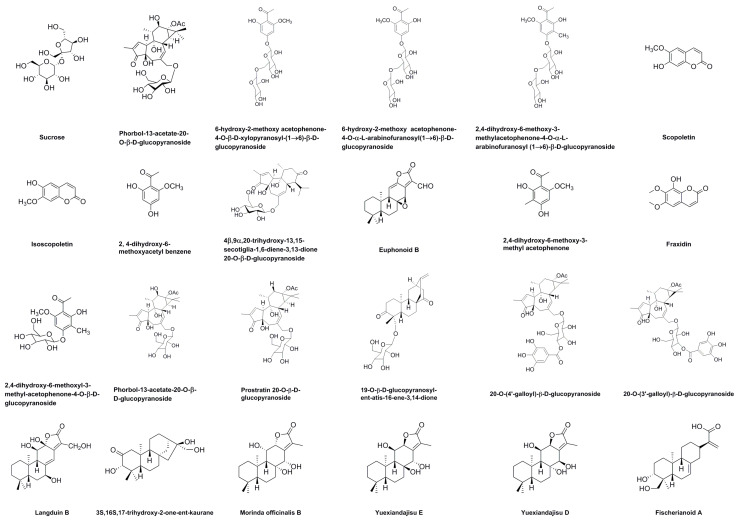
Chemical structures of the compounds identified in acetone extract of *Euphorbia fischeriana*.

**Figure 5 molecules-28-05172-f005:**
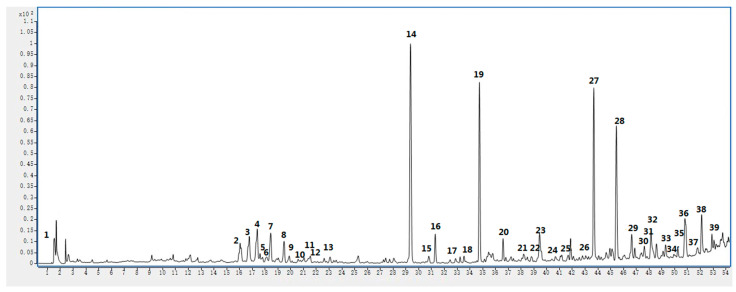
UHPLC–MS results captured in positive-ion mode for acetone extract of *Euphorbia fischeriana*.

**Figure 6 molecules-28-05172-f006:**
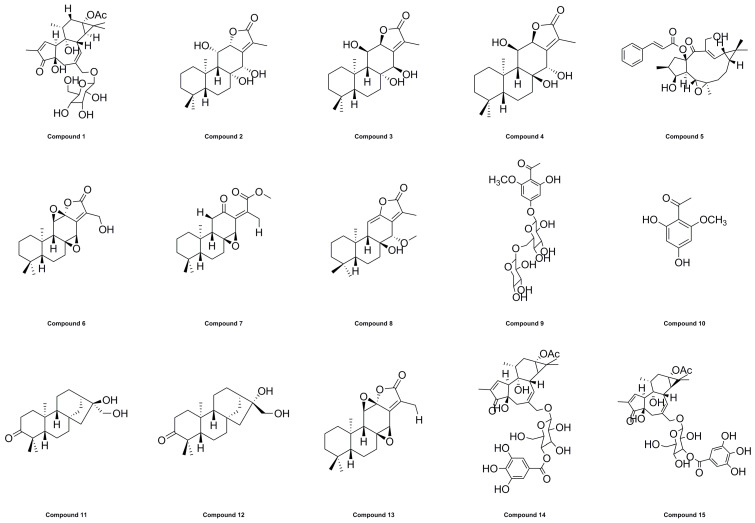
Chemical structures of compounds **1**–**15**.

**Figure 7 molecules-28-05172-f007:**
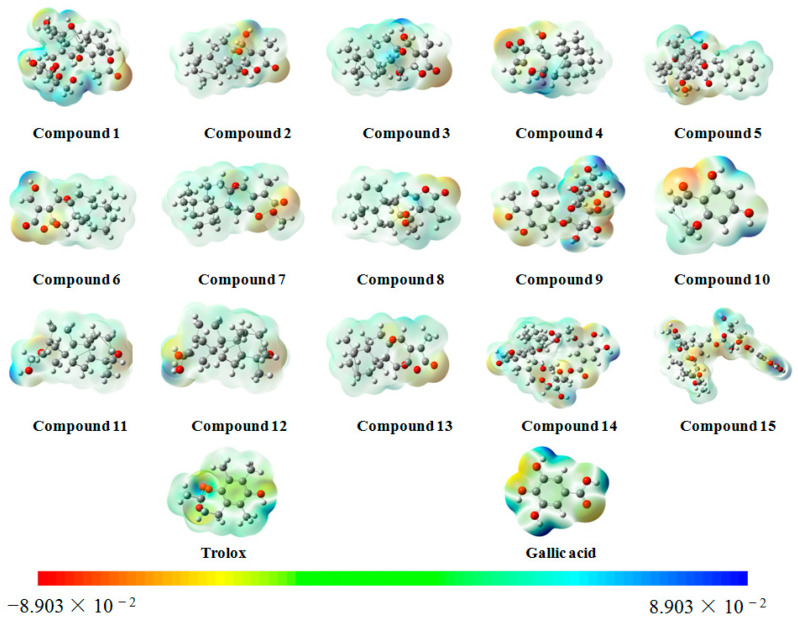
MEP surface map of compounds **1**–**15**.

**Figure 8 molecules-28-05172-f008:**
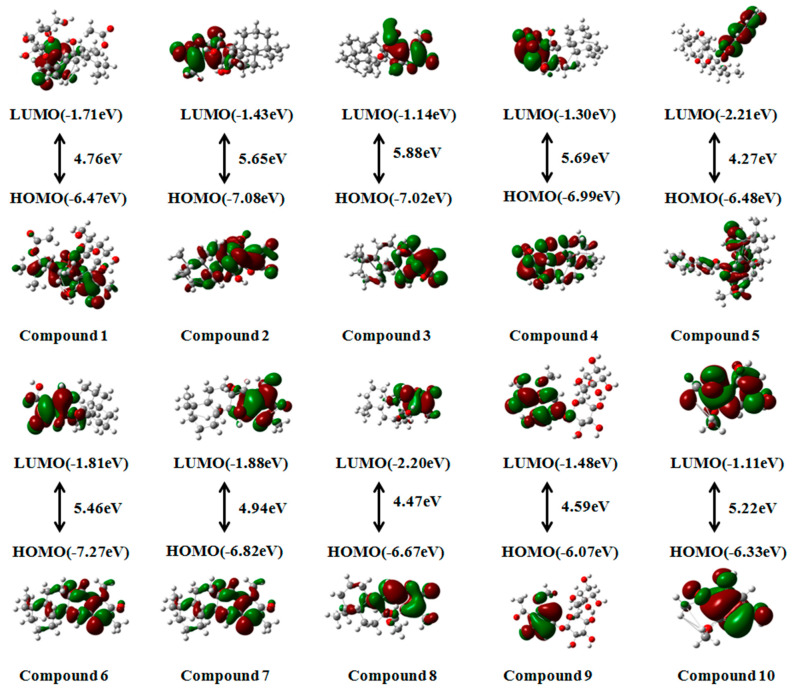
Frontier molecular orbitals of compounds **1**–**15**.

**Figure 9 molecules-28-05172-f009:**
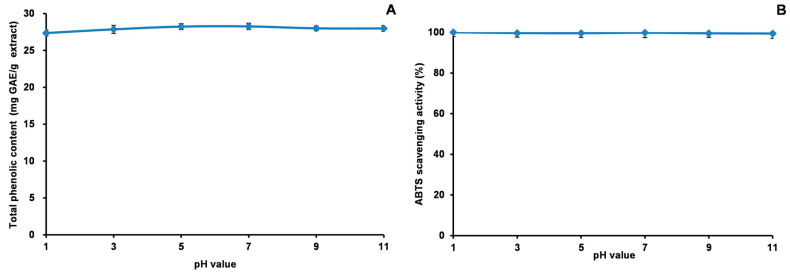
Total phenolic content (**A**) and ABTS (**B**) assays to assess the stability of the acetone extract of *Euphorbia fischeriana* at various pH values (ABTS: 2,2′-Azino-bis (3-ethylbenzothiazoline-6-sulphonic acid) diammonium salt).

**Figure 10 molecules-28-05172-f010:**
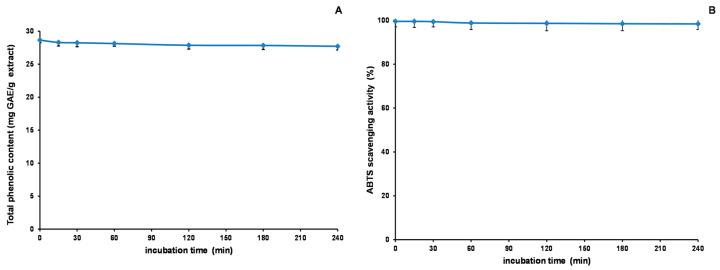
Total phenolic content (**A**) and ABTS (**B**) assays to assess the thermal stability of the acetone extract of *Euphorbia fischeriana* (ABTS: 2,2′-Azino-bis (3-ethylbenzothiazoline-6-sulphonic acid) diammonium salt).

**Figure 11 molecules-28-05172-f011:**
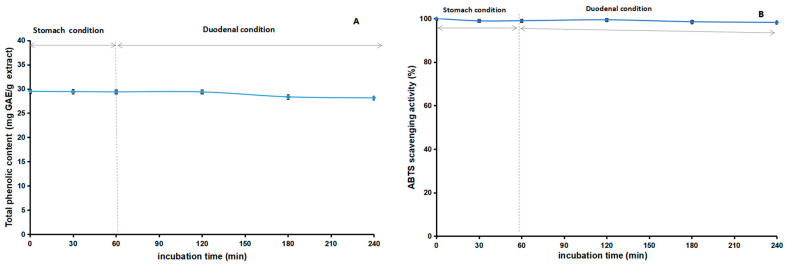
Total phenolic content (**A**) and ABTS (**B**) assays to assess the stability of the acetone extract of *Euphorbia fischeriana* in vitro simulation of the human digestive system (ABTS: 2,2′-Azino-bis (3-ethylbenzothiazoline-6-sulphonic acid) diammonium salt).

**Figure 12 molecules-28-05172-f012:**
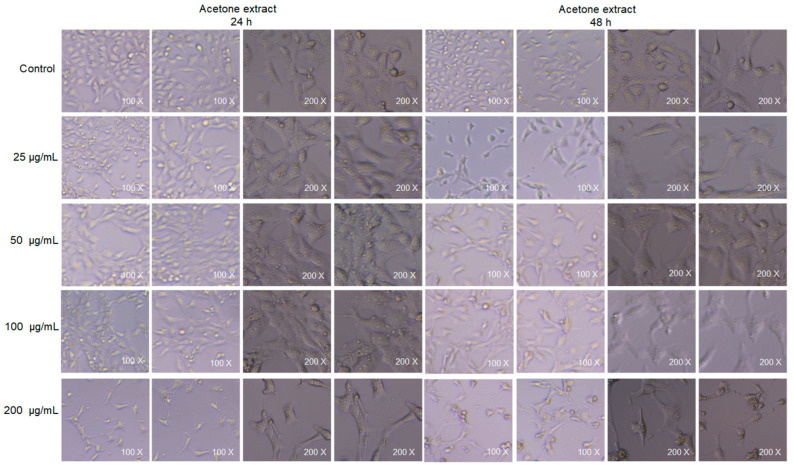
Morphology of TM_3_ mouse cells treated with the acetone extract of *Euphorbia fischeriana* for 24 or 48 h.

**Figure 13 molecules-28-05172-f013:**
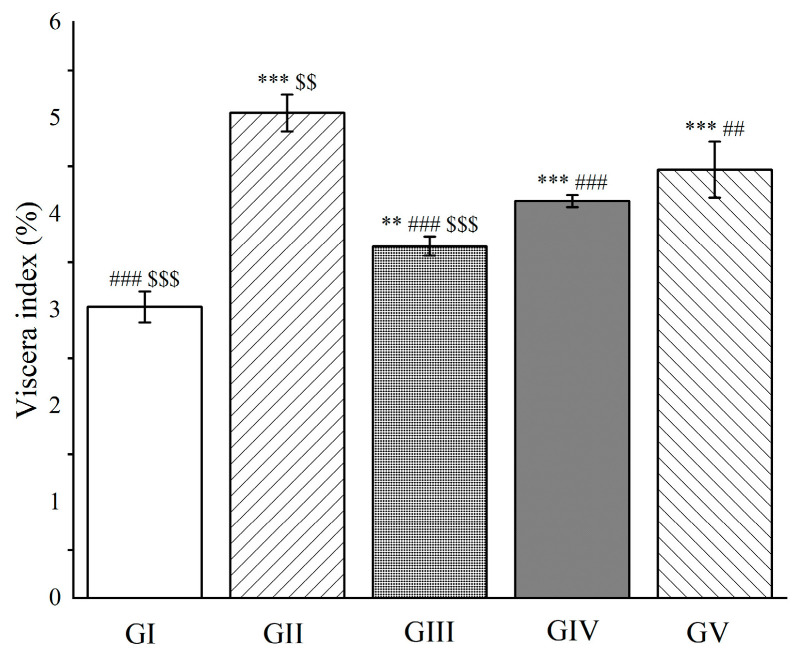
The outcomes of treatment with a *Euphorbia fischeriana* acetone extract on the hepatic viscera index in rats with liver injury. Values are expressed as the mean ± standard error of the mean (n = 8). GI: Control group, GII: D-GalN group, GIII: D-GalN + EF_300_ group, GIV: D-GalN + EF_150_ group, GIV: D-GalN + SMN group. EF: *Euphorbia fischeriana*; D-GalN: D-Galactosamine; SMN: Silymarin. Significantly different from the control group at ** *p* < 0.01 and *** *p* < 0.001. Significantly different from the D-GalN group at ## *p* < 0.01 and ### *p* < 0.001. Significantly different from the D-GalN + SMN group at $$ *p* < 0.01 and $$$ *p* < 0.001.

**Figure 14 molecules-28-05172-f014:**
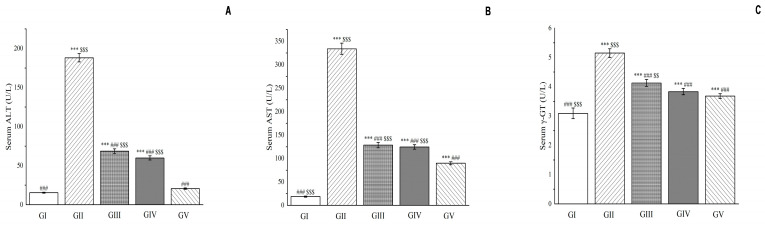
Effects of *Euphorbia fischeriana* on serum alanine aminotransferase (**A**), aspartate aminotransferase (**B**), and γ-GT (**C**) in rats with liver injury. Values are expressed as the mean ± standard error of the mean (*n* = 8). GI: Control group, GII: D-GalN group, GIII: D-GalN + EF_300_ group, GIV: D-GalN + EF_150_ group, GIV: D-GalN + SMN group. EF: *Euphorbia fischeriana*; D-GalN: D-Galactosamine; SMN: Silymarin; ALT: Alanine aminotransferase; AST: Aspartate aminotransferase; γ-GT: γ-Glutamyl transpeptidase. Significantly different from the control group at *** *p* < 0.001. Significantly different from the D-GalN group at ### *p* < 0.001. Significantly different from the D-GalN + SMN group at $$ *p* < 0.01 and $$$ *p* < 0.001.

**Figure 15 molecules-28-05172-f015:**
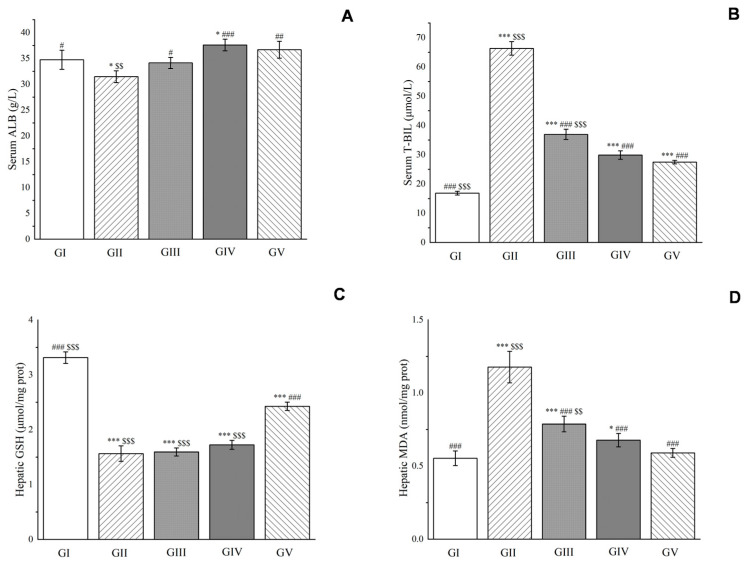
Effects of *Euphorbia fischeriana* on serum albumin (**A**), total bilirubin (**B**), hepatic glutathione (**C**), and malondialdehyde (**D**) in rats with liver injury. Values are expressed as the mean ± standard error of the mean (n = 8). GI: Control group, GII: D-GalN group, GIII: D-GalN + EF_300_ group, GIV: D-GalN + EF_150_ group, GIV: D-GalN + SMN group. EF: *Euphorbia fischeriana*; D-GalN: D-Galactosamine; SMN: Silymarin; ALB: Albumin; T-BIL: Total bilirubin; GSH: Glutathione; MDA: Malondialdehyde. Significantly different from the control group at * *p* < 0.05 and *** *p* < 0.001. Significantly different from the D-GalN group at # *p* < 0.05 ## *p* < 0.01 and ### *p* < 0.001. Significantly different from the D-GalN + SMN group at $$ *p* < 0.01 and $$$ *p* < 0.001.

**Figure 16 molecules-28-05172-f016:**
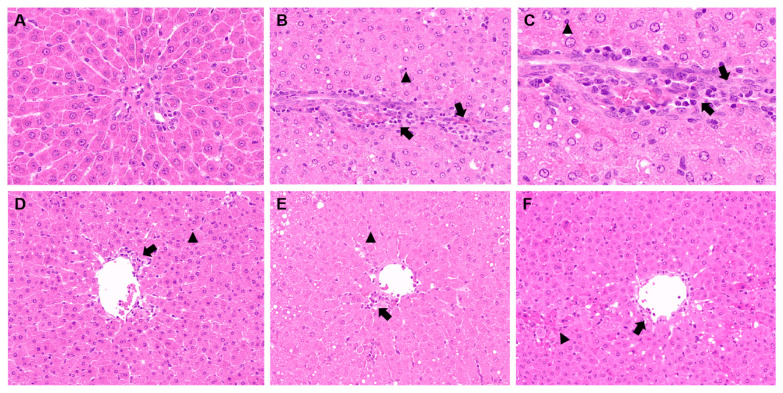
Histological examination of liver sections in different groups (200× magnification). (**A**) Control group (200× magnification); (**B**) D-GalN group (200× magnification); (**C**): D-GalN group (400× magnification); (**D**) D-GalN + EF_300_ group (200× magnification); (**E**) D-GalN + EF_150_ group (200× magnification); (**F**) D-GalN + SMN group (200× magnification). EF: *Euphorbia fischeriana*; D-GalN: D-Galactosamine; SMN: Silymarin. No-tailed arrow: single-cell necrosis; Long-tailed arrow: inflammatory cells.

**Table 1 molecules-28-05172-t001:** Phytochemical analysis of *Euphorbia fischeriana*.

Phytochemicals	Type of Tests	Sample Solution		
		Water	Methanol	Petroleum Ether
Proteins/amino acids	1. Ninhydrin tests	+	○	○
	2. Biuret tests	+	○	○
Carbohydrates	1. Fehling’s tests	+	○	○
	2. Benedict’s tests	+	○	○
	3. Molisch’s tests	+	○	○
	4. Iodine tests	+	○	○
Phenols	1. FeCl_3_ tests	+	○	○
	2. FeCl_3_-K_3_[Fe(CN)_6_] tests	+	○	○
	3. Diazotization tests	+	○	○
Organic acids	1. pH tests	+	○	○
	2. Blue litmus paper tests	+	○	○
	3. Bromocresol green tests	+	○	○
Tannins	1. FeCl_3_ tests	+	○	○
	2. Bromine water tests	+	○	○
	3. Lead acetate tests	+	○	○
	4. Lime water tests	+	○	○
	5. Gelatin tests	+	○	○
Flavonoids	1. Shinoda tests	○	−	○
	2. Alkaline reagent tests	○	−	○
	3. AlCl_3_ tests	○	+	○
	4. Lead acetate tests	○	+	○
Saponins	1. Foam tests	−	○	○
Steroids and triterpenoids	1. Liebermann–Burchard tests	○	+	○
	2. Salkowski tests	○	+	○
Terpenoids	1. CHCl_3_-H_2_SO_4_ tests	○	+	○
	2. Vanillin-H_2_SO_4_ tests	○	○	+
Alkaloids	1. Bertrad’s reagent	○	+	○
	2. Dragendorff’s reagent	○	+	○
	3. Mayer’s reagent	○	+	○
Anthraquinones	1. Borntrager’s tests	○	−	○
	2. Magnesium acetate tests	○	−	○
Coumarins and lactones	1. Hydroxamic acid iron tests	+	○	○
	2. Diazotization tests	+	○	○
	3. Fluorescence tests	○	+	○
Volatile oils and fats	1. Phosphomolybdic acid tests	○	+	○
	2. Vanillin-H_2_SO_4_ tests	○	+	○
	3. Sudan tests	○	+	○
Cardiac glycosides	1. Kedde tests	○	−	○
	2. Raymond tests	○	−	○
	3. Legal tests	○	−	○
Cyanogenic glycosides	1. Prussian blue tests	−	○	○

(+) indicates presence; (−) indicates absence; (○) indicates no test.

**Table 2 molecules-28-05172-t002:** Extraction yields of *Euphorbia fischeriana* extracted with different solvents.

Extracting Solvents	Yields (%, *w*/*w*)
Water	33.4 ± 0.4 ^a^
Methanol	25.1 ± 0.6 ^b^
Ethanol	20.4 ± 0.4 ^c^
Acetone	13.3 ± 0.1 ^d^
Ethyl acetate	12.3 ± 0.2 ^e^
Ethyl ether	10.2 ± 0.1 ^f^
Dichloromethane	8.0 ± 0.1 ^g^
Hexane	8.1 ± 0.4 ^g^

^a–g^ Columns with different superscripts indicate a significant difference (*p* < 0.05).

**Table 3 molecules-28-05172-t003:** Total carbohydrate content (TCC), total protein content (TP_ro_C), total triterpenoid content (TT_ri_C), total phenolic content (TP_he_C), total flavonoid content (TFC), total tannin content (TT_an_C), gallotannin content (GC), condensed tannin content (CTC), and total alkaloid content (TAC) of *Euphorbia fischeriana* extracted with different solvents.

Extracting Solvents	TCC (mg GE/g Extract)	TP_ro_C (mg BSAE/g Extract)	TT_ri_C (mg GRE/g Extract)	TP_he_C (mg GAE/g Extract)	TFC (mg QE/g Extract)	TT_an_C (mg TAE/g Extract)	GC (mg GAE/g Extract)	CTC (mg GAE/g Extract)	TAC (mg BHE/g Extract)
Water	537.8 ± 19.6 ^a^	643.2 ± 12.5 ^a^	NONE	37.3 ± 2.3 ^a^	1.4 ± 0.1 ^c^	36.9 ± 1.2 ^a^	30.4 ± 1.4 ^d^	23.0 ± 0.5 ^c^	1.9 ± 0.0 ^b^
Methanol	263.9 ± 8.1 ^c^	110.5 ± 2.6 ^d^	362.5 ± 15.5 ^d^	11.6 ± 0.3 ^c^	2.4 ± 0.3 ^b^	11.2 ± 0.2 ^d^	97.5 ± 2.1 ^b^	18.6 ± 0.2 ^d^	2.4 ± 0.0 ^a^
Ethanol	501.9 ± 8.7 ^b^	489.7 ± 3.1 ^b^	NONE	29.7 ± 1.2 ^b^	3.2 ± 0.1 ^a^	28.5 ± 0.4 ^c^	113.3 ± 3.3 ^a^	27.1 ± 0.1 ^b^	1.9 ± 0.0 ^a^
Acetone	76.2 ± 1.6 ^e^	441.1 ± 5.8 ^c^	870.5 ± 49.8 ^a^	38.6 ± 2.2 ^a^	2.7 ± 0.2 ^b^	32.7 ± 1.3 ^b^	115.9 ± 5.1 ^a^	35.1 ± 0.4 ^a^	1.8 ± 0.0 ^a^
Ethyl acetate	101.9 ± 4.8 ^d^	NONE	632.9 ± 30.2 ^b^	5.7 ± 0.3 ^d^	2.1 ± 0.1 ^c^	5.3 ± 0.1 ^e^	47.5 ± 1.7 ^c^	8.7 ± 0.1 ^e^	1.7 ± 0.0 ^c^
Ethyl ether	0.4 ± 0.0 ^f^	NONE	NONE	NONE	1.5 ± 0.0 ^c^	0.3 ± 0.0 ^f^	10.3 ± 0.5 ^e^	2.1 ± 0.1 ^f^	1.7 ± 0.1 ^a^
Dichloromethane	NONE	NONE	499.0 ± 28.0 ^c^	0.3 ± 0.0 ^e^	1.6 ± 0.0 ^c^	0.6 ± 0.0 ^f^	10.9 ± 0.5 ^e^	1.8 ± 0.1 ^f^	1.6 ± 0.0 ^a^
Hexane	2.3 ± 0.1 ^f^	NONE	529.0 ± 15.0 ^c^	NONE	1.4 ± 0.0 ^c^	NONE	7.6 ± 0.2 ^e^	0.7 ± 0.0 ^g^	1.7 ± 0.0 ^c^

^a–g^ Columns with different superscripts indicate a significant difference (*p* < 0.05).

**Table 4 molecules-28-05172-t004:** Determination of antioxidant activity of various solvent extracts of *Euphorbia fischeriana* using DPPH, ABTS, hydroxyl, and superoxide radicals.

Extracting Solvents	DPPH (mg TE/g Extract)	ABTS (mg TE/g Extract)	Hydroxyl Radicals (mg TE/g Extract)	Superoxide Radicals (%, 2143 μg/mL)
Water	51.1 ± 1.6 ^e^	119.2 ± 4.4 ^c^	75.1 ± 3.1 ^e^	25.1 ± 1.0 ^d^
Methanol	391.2 ± 11.7 ^a^	98.2 ± 3.7 ^d^	184.2 ± 6.4 ^c^	28.1 ± 1.1 ^d^
Ethanol	334.7 ± 13.4 ^b^	173.3 ± 8.1 ^b^	283.4 ± 11.0 ^b^	28.1 ± 1.0 ^d^
Acetone	264.2± 10.6 ^c^	240.2 ± 10.1 ^a^	321.1 ± 12.9 ^c^	12.3 ± 0.3 ^f^
Ethyl acetate	109.1 ± 4.5 ^d^	80.4 ± 3.0 ^e^	97.4 ± 3.8 ^d^	22.9 ± 0.8 ^e^
Ethyl ether	28.2 ± 1.0 ^f^	28.4 ± 0.8 ^f^	<44.1 ^f^	44.5 ± 1.6 ^b^
Dichloromethane	20.2 ± 0.7 ^f^	32.4 ± 1.1 ^f^	76.6 ± 3.6 ^e^	47.5 ± 1.8 ^b^
Hexane	15.1 ± 0.5 ^f^	8.1 ± 0.3 ^g^	<44.1 ^f^	39.0 ± 1.4 ^c^
Curcumin *	N.T.	N.T.	N.T.	60.3 ± 1.0 ^a^

^a–g^ Columns with different superscripts indicate a significant difference (*p* < 0.05). * Used as a standard antioxidant. N.T. indicates no test.

**Table 5 molecules-28-05172-t005:** Determination of antioxidant activity of various solvent extracts of *Euphorbia fischeriana* using FRAP, CUPRAC, and metal chelating.

Extracting Solvents	FRAP (mg TE/g Extract)	CUPRAC (mg TE/g Extract)	Iron Chelating (mg EDTAE/g Extract)	Copper Chelating (mg EDTAE/g Extract)
Water	458.3 ± 13.1 ^b^	60.2 ± 1.8 ^e^	<1.1 ^d^	88.0 ± 2.7 ^a^
Methanol	750.0 ± 20.5 ^a^	313.3 ± 10.9 ^c^	4.2 ± 0.1 ^a^	29.1 ± 0.9 ^d^
Ethanol	750.0 ± 19.8 ^a^	349.4 ± 10.5 ^b^	<1.1 ^d^	61.7 ± 1.9 ^c^
Acetone	750.0 ± 19.4 ^a^	373.5 ± 11.3 ^a^	1.6 ± 0.0 ^c^	49.7 ± 1.3 ^b^
Ethyl acetate	458.3 ± 13.4 ^b^	132.5 ± 4.0 ^d^	4.3 ± 0.1 ^a^	150.6 ± 4.6 ^e^
Ethyl ether	333.3 ± 11.9 ^c^	NONE	1.9 ± 0.1 ^b^	<10.40 ^f^
Dichloromethane	333.3 ± 12.1 ^c^	NONE	<1.1 ^d^	<10.40 ^f^
Hexane	333.3 ± 12.4 ^c^	NONE	<1.1 ^d^	<10.40 ^f^

^a–f^ Columns with different superscripts indicate a significant difference (*p* < 0.05).

**Table 6 molecules-28-05172-t006:** Determination of antioxidant activity of various solvent extracts of *Euphorbia fischeriana* using H_2_O_2_, β-carotene bleaching, singlet oxygen, and HClO.

Extracting Solvents	H_2_O_2_ (mg GAE/g Extract)	β-Carotene Bleaching AAC	Singlet Oxygen (%, 2000 μg/mL)	HClO (mg TE/g Extract)
Water	18.1 ± 0.5 ^b^	662.6 ± 24.4 ^c^	18.7 ± 0.6 ^g^	93.1 ± 3.4 ^a^
Methanol	12.1 ± 0.4 ^c^	894.7 ± 38.4 ^a^	30.6 ± 0.9 ^b^	24.2 ± 0.7 ^d^
Ethanol	9.2 ± 0.3 ^d^	863.9 ± 38.4 ^a^	22.1 ± 0.8 ^e^	47.2 ± 1.9 ^c^
Acetone	53.1 ± 1.7 ^a^	864.3 ± 42.0 ^a^	25.4 ± 0.7 ^c^	53.1 ± 2.3 ^b^
Ethyl acetate	<6.0 ^e^	748.5 ± 34.8 ^b^	23.9 ± 1.5 ^d^	16.1 ± 0.4 ^e^
Ethyl ether	<6.0 ^e^	513.2 ± 26.9 ^d^	23.8 ± 0.9 ^d^	<12.8 ^f^
Dichloromethane	<6.0 ^e^	668.0 ± 32.3 ^c^	28.3 ± 0.8 ^b^	<12.8 ^f^
Hexane	<6.0 ^e^	733. 3 ± 30.1 ^b^	19.1 ± 0.6 ^f^	<12.8 ^f^
BHT *	N.T.	908.4 ± 46.5 ^a^	N.T.	N.T.
BHA *	N.T.	901.9 ± 45.0 ^a^	N.T.	N.T.
Ferulic acid *	N.T.	N.T.	95.3 ± 3.2 ^a^	N.T.

^a–g^ Columns with different superscripts indicate a significant difference (*p* < 0.05). * Used as a standard antioxidant. N.T. indicates no test.

**Table 7 molecules-28-05172-t007:** Compounds identified in acetone extract of *Euphorbia fischeriana*.

Peak No.	RT (min)	Identification	Molecular Formula	Selective Ion	Full Scan MS (*m*/*z*)		Error (ppm)	MS/MS Fragments (*m*/*z*)
					Theory	Measured		
**1**	1.53	Sucrose	C_12_H_22_O_11_	[M + Na]^+^	365.1060	365.1062	−0.5	162.1125
**2**	16.04	Phorbol-13-acetate-20-*O*-β-d-glucopyranoside	C_28_H_40_O_12_	[M + H]^+^	569.2598	569.2595	0.5	311.1644, 293.1538
**3**	16.77	6-Hydroxy-2-methoxy acetophenone-4-*O*-β-d-xylopyranosyl-(1→6)-β-d-glucopyranoside	C_20_H_28_O_13_	[M + H]^+^	477.1608	477.1616	−1.7	345.1189, 183.0654
**4**	17.38	6-Hydroxy-2-methoxy acetophenone-4-*O*-α-L-arabinofuranosyl-(1→6)-β-d-glucopyranoside	C_20_H_28_O_13_	[M + H]^+^	477.1608	477.1617	−1.9	345.1196, 183.0660, 165.0551
**5**	17.79	2,4-Dihydroxy-6-methoxy-3-methylacetophenone-4-*O*-α-L-arabinofuranosyl-(1→6)-β-d-glucopyranoside	C_21_H_30_O_13_	[M + H]^+^	491.1765	491.1761	0.8	345.1188, 183.0654
**6**	18.10	Scopoletin/Isoscopoletin	C_10_H_8_O_4_	[M + H]^+^	193.0501	193.0497	2.1	121.0289
**7**	18.43	2, 4-Dihydroxy-6-methoxyacetyl benzene	C_9_H_10_O_4_	[M + H]^+^	183.0657	183.0654	1.6	165.0550, 153.0184
**8**	19.48	4β,9α,20-Trihydroxy-13,15-secotiglia-1,6-diene-3,13-dione 20-*O*-β-d-glucopyranoside	C_26_H_38_O_10_	[M + Na]^+^	533.2363	533.2365	−0.4	295.1694
**9**	19.87	Euphonoid B	C_20_H_24_O_4_	[M + H]^+^	329.1753	329.1756	−0.9	313.1725
**10**	20.57	2,4-Dihydroxy-6-methoxy-3-methyl acetophenone	C_10_H_12_O_4_	[M + H]^+^	197.0814	197.0813	0.5	179.0705, 165.0542, 153.0189
**11**	20.93	Fraxidin	C_11_H_10_O_5_	[M + H]^+^	223.0606	223.0601	2.2	208.0361, 179.0701
**12**	21.05	2,4-Dihydroxy-6-methoxyl-3-methyl-acetophenone-4-*O*-β-d-glucopyranoside	C_16_H_22_O_9_	[M + H]^+^	359.1342	359.1340	0.6	315.0715, 197.0809, 179.0703
**13**	22.60	Phorbol-13-acetate-20-*O*-β-d-glucopyranoside	C_28_H_40_O_12_	[M + H]^+^	569.2598	569.2597	0.2	591.2413, 511.1779
**14**	29.35	Prostratin 20-*O*-β-d-glucopyranoside	C_28_H_40_O_11_	[M + H]^+^	553.2649	553.2663	−2.5	277.1587
**15**	30.80	19-*O*-β-d-Glucopyranosyl-ent-atis-16-ene-3,14-dione	C_26_H_38_O_8_	[M + Na]^+^	501.2464	501.2465	−0.2	317.2113, 299.2002
**16**	31.29	20-*O*-(4′-Galloyl)-β-d-glucopyranoside/20-*O*-(3′-Galloyl)-β-d-glucopyranoside	C_35_H_44_O_15_	[M + H]^+^	705.2758	705.2777	−2.7	1426.5730, 153.0185
**17**	32.45	Langduin B	C_20_H_28_O_6_	[M + H]^+^	365.1964	365.1961	0.8	347.1861, 329.1754
**18**	32.88	3S,16S,17-Trihydroxy-2-one-ent-kaurane	C_20_H_32_O_4_	[M + H]^+^	337.2379	337.2381	−0.6	359.2201, 319.2276, 277.1590
**19**	34.75	Morinda officinalis B/Yuexiandajisu E/Yuexiandajisu D	C_20_H_30_O_5_	[M + Na]^+^	373.1991	373.2025	−9.11	295.1702
**20**	36.59	Unknown				364.2483		148.0756, 131.0492, 105.0697
**21**	38.20	Fischerianoid A	C_20_H_30_O_4_	[M + H]^+^	335.2222	335.2217	1.5	357.2050, 317.2116,
**22**	38.80	Ent-(13S)-13-hydroxyatis-16-ene-3,14-dione	C_20_H_28_O_3_	[M + H]^+^	317.2117	317.2112	1.6	339.1936, 289.2167, 261.1852
**23**	39.44	Ent-kaurane-3-oxo-16α,17-diol/Ent-kaurane-3-oxo-16β,17-diol	C_20_H_32_O_3_	[M + Na]^+^	343.2249	343.2254	−1.5	321.2433, 303.2327
**24**	40.69	Fischeriabietane E	C_21_H_30_O_6_	[M + H]^+^	379.2121	379.2120	0.3	361.2024, 333.2067,319.2278
**25**	41.67	Araucarone	C_20_H_30_O_3_	[M + H]^+^	319.2273	319.2265	2.5	181.1018
**26**	42.79	7β,11β,12β-Trihydroxy-ent abieta-8(14),13(15)-dien-16,12-olide	C_20_H_28_O_5_	[M + H]^+^	349.2015	349.2011	1.1	367.1529, 313.1809, 331.1913
**27**	43.68	Jolkinol A	C_29_H_36_O_6_	[M + Na]^+^	503.2410	503.2412	−0.4	481.2593, 463.2489, 131.0492
**28**	45.44	17-Hydroxyjolkinolide B	C_20_H_26_O_5_	[M + Na]^+^	369.1678	369.1675	0.8	—
**29**	46.65	13β-Hydroxy-7-oxobiet-8(14)-en-19,6β-olide	C_20_H_28_O_4_	[M + H]^+^	333.2066	333.2070	−1.2	315.1961, 287.2374
**30**	46.88	Jolkinol B	C_29_H_36_O_5_	[M + Na]^+^	487.2460	487.2457	0.6	447.2540, 419.2247
**31**	47.36	Fischeriana A	C_27_H_30_O_8_	[M + Na]^+^	505.1838	505.1839	−0.2	483.2018, 331.1813
**32**	48.16	Jolkinolide B	C_20_H_26_O_4_	[M + Na]^+^	353.1729	353.1726	0.8	331.1912, 303.2324
**33**	49.29	Ent-13-hydroxyatis-16-ene-3,14-dione	C_20_H_28_O_3_	[M + H]^+^	317.2117	317.2117	0.0	339.1935, 299.2013
**34**	49.95	Ent-atis-16(17)-ene-3,14-dione/Ent-atis-16-ene-3,14-dione	C_20_H_28_O_2_	[M + H]^+^	301.2168	301.2162	2.0	285.2216, 271.2425
**35**	50.24	Fischernolide D	C_29_H_32_O_8_	[M + H]^+^	509.2175	509.2170	1.0	491.2072, 463.2122
**36**	50.80	Unknown				331.1913		317.2119, 299.2029, 277.2163
**37**	51.78	Ent-10α-hydroxy-rosa-1,15-dien-3-one	C_20_H_30_O_2_	[M + H]^+^	303.2324	303.2318	2.0	285.2221, 257.2264,
**38**	52.11	Euphonoid A	C_21_H_30_O_4_	[M + H]^+^	347.2222	347.2222	0.0	298.1891, 269.1904
**39**	52.91	Landuin D	C_29_H_32_O_9_	[M + H]^+^	525.2125	525.2126	−0.2	481.2230

RT: Retention time.

**Table 8 molecules-28-05172-t008:** Number and name of compounds corresponding to ten peaks in ion flow diagram.

Peak No.	Compound No.	Identification
14	**1**	Prostratin 20-*O*-β-d-glucopyranoside
19	**2**	Morinda officinalis B
	**3**	Yuexiandajisu E
	**4**	Yuexiandajisu D
27	**5**	Jolkinol A
28	**6**	17-Hydroxyjolkinolide B
38	**7**	Fischeriabietane B
	**8**	Euphonoid A
4	**9**	6-Hydroxy-2-methoxy 4-*O*-α-L-arabinofuranosyl(1→6)-β-d-glucopyranoside
7	**10**	2,4-Dihydroxy-6-methoxyacetyl benzene
23	**11**	Ent-kaurane-3-oxo-16α,17-diol
	**12**	Ent-kaurane-3-oxo-16β,17-diol
32	**13**	Jolkinolide B
16	**14**	20-*O*-(4′-Galloyl)-β-d-glucopyranoside
	**15**	20-*O*-(3′-Galloyl)-β-d-glucopyranoside

**Table 9 molecules-28-05172-t009:** Determination of cytotoxicity of the acetone extract of *Euphorbia fischeriana* by the MTT method.

Acetone Extract (μg/mL)	Cell Survival Rate of TM_3_ Cells (%)	
	24 h	48 h
0	100.00 ± 2.41 ^a^	100.00 ± 2.34 ^b^
25	102.29 ± 1.73 ^a^	116.94 ± 3.25 ^a^
50	102.98 ± 2.35 ^a^	118.65 ± 4.30 ^a^
100	86.95 ± 1.49 ^b^	86.67 ± 1.99 ^c^
200	75.25 ± 2.30 ^c^	37.11 ± 1.34 ^d^

^a–d^ Columns with different superscripts indicate a significant difference (*p* < 0.05).

## Data Availability

All data presented in this study are available in the article.

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
