# Peer review of "Phytochemical Analysis, Antioxidant Activities In Vitro and In Vivo, and Theoretical Calculation of Different Extracts of Euphorbia fischeriana"

_molecules, 2023, doi:10.3390/molecules28135172_

Round 1

Reviewer 1 Report

In the paper entitled "Phytochemical analysis, antioxidant activities in vitro, vivo and theoretical calculation of different extracts of Euphorbia fischeriana " by Sun et al.,, the antioxidant properties of Euphorbia fischeriana were investigated. UV-Vis spectrophotometry was used to evaluate the antioxidant properties of eight different solvent extracts. Only acetone extract was selected for further study. Utilising UV-Vis spectrophotometry, the antioxidant properties of eight distinct solvent extracts were determined. Only the acetone extract was chosen for further study.

The UHPLC-MS method was used to identify the corresponding compounds in the acetone extract. Several experiments were conducted to determine the stability and acute oral toxicity of the extract. This study is very interesting and gives good insight into the antioxidant properties of this plant. However, some corrections need to be made.

- Since the ethanol extract exhibited comparable activity to the acetone extract, I recommend analysing and comparing this extract with the acetone extract.

The experimental part of determining antioxidant activity is inadequately described. This study determined the Trolox equivalent antioxidant capacity (TEAC). Therefore, it is necessary to briefly describe these tests and provide Trolox standard curves for all tests of antioxidant activity where results are expressed in this manner in the supplementary material.

- Discussion in chapter 2.5. Frontier molecular orbital should be written better. See the following references: doi /10.1039/c5ra02134k, doi /10.1016/j.proci.2022.07.168, doi /10.1016/j.molstruc.2020.127935. The obtained results should be compared with the existing ones. This is one of the main shortcomings of manuscripts.

- In tables 3 and 7, it is not possible to see all the displayed results.

Author Response

Question 1: In the paper entitled "Phytochemical analysis, antioxidant activities in vitro, vivo and theoretical calculation of different extracts of Euphorbia fischeriana " by Sun et al., the antioxidant properties of Euphorbia fischeriana were investigated. UV-Vis spectrophotometry was used to evaluate the antioxidant properties of eight different solvent extracts. Only acetone extract was selected for further study. Utilising UV-Vis spectrophotometry, the antioxidant properties of eight distinct solvent extracts were determined. Only the acetone extract was chosen for further study.

The UHPLC-MS method was used to identify the corresponding compounds in the acetone extract. Several experiments were conducted to determine the stability and acute oral toxicity of the extract. This study is very interesting and gives good insight into the antioxidant properties of this plant. However, some corrections need to be made.

Answer: Dear reviewer, thank you very much for your recognition of our work. And thank you for your valuable comments to help us improve our manuscript. We have answered your concerns and revised the questions in our manuscript according to your requirements, after revision, the quality of our manuscript has been improved.

Question 2: - Since the ethanol extract exhibited comparable activity to the acetone extract, I recommend analysing and comparing this extract with the acetone extract.

Answer: Thank you very much for your advice! We have analyzed and compared ethanol extract and acetone extract in our manuscript.

Question 3: The experimental part of determining antioxidant activity is inadequately described. This study determined the Trolox equivalent antioxidant capacity (TEAC). Therefore, it is necessary to briefly describe these tests and provide Trolox standard curves for all tests of antioxidant activity where results are expressed in this manner in the supplementary material.

Answer: Your question is very professional. We have added the methods of all antioxidant experiments and the standard curves of standard antioxidants (not only trolox, but also conclude antioxidants such as ferulic acid and gallic acid) to the supplementary material. After revision according to your requirements, our manuscript would have more reference value and be more helpful to readers.

Question 4: - Discussion in chapter 2.5. Frontier molecular orbital should be written better. See the following references: doi /10.1039/c5ra02134k, doi /10.1016/j.proci.2022.07.168, doi /10.1016/j.molstruc.2020.127935. The obtained results should be compared with the existing ones. This is one of the main shortcomings of manuscripts.

Answer: Your suggestion is very good. We have read the three references recommended by you and have benefited a lot. We also cited them in our manuscript.

However, one thing to explain is that the activity evaluation experiment of this study is to do the antioxidant activity of different solvent extracts, rather than the antioxidant activity of pure compounds. Therefore, there is no experimental result of fifteen compounds identified by UHPLC-MS, moreover, there are no reports of antioxidant activity in these compounds for reference, so the calculated results cannot be compared with the experimental results. The revised manuscript could provide more useful information about frontier molecular orbital, the discussion is more in-depth. Thank you very much!

Question 5: - In tables 3 and 7, it is not possible to see all the displayed results.

Answer: Your reminder is very important. We have adjusted the table 3 and table 7 according to your requirements. Now, all the results in table 3 and table 7 could be seen.

Reviewer 2 Report

1. Phenolic compounds, according to the results of the authors (Table 1), are best extracted by water, while it is well known that this class of compounds has high antioxidant activity (AA), most likely not high indicators in the tests are associated with ballast substances in an aqueous extract, however, in my opinion, water and water-alcohol extracts should be studied more closely.

2. Tables 3 and 7 are incorrectly displayed

3. Experiment planning should be improved and made more consistent. So, for example, the authors chose 8 solvents for the work, but the determination of the classes of compounds was carried out only for 3, not in all tests acetone extracts showed the best results, while LC-MS was carried out only for him.  According to the authors, the best solvent is acetone, which is absent in Table 1.

Author Response

Question 1: Phenolic compounds, according to the results of the authors (Table 1), are best extracted by water, while it is well known that this class of compounds has high antioxidant activity (AA), most likely not high indicators in the tests are associated with ballast substances in an aqueous extract, however, in my opinion, water and water-alcohol extracts should be studied more closely.

Answer: Dear reviewer, thank you very much for your recognition of our work. And thank you for your valuable comments to help us improve our manuscript. We have answered your concerns and revised the questions in our manuscript according to your requirements, after revision, the quality of our manuscript has been improved. We are sorry that we have not written down the specific experimental methods in our manuscript, which has led to your misunderstanding. Now, we have listed all the experimental methods in the supplementary material. Table 1 shows the experimental results of qualitative experiments. Aqueous extraction solution is used to check for carbohydrates, organic acids, saponins, glycosides, phenols, tannins and cyanogenic glycosides.These types of phytochemicals have high polarity and could be dissolved by water. Methanol extraction solution is used to check for flavonoids, anthraquinones, cardiac glycosides, coumarins, lactones, volatile oils, terpenoids, steroids and lipids.These types of phytochemicals have relative high polarity and could be dissolved by methanol. Petroleum ether extraction solution is used to check for volatile oils and lipids, steroids or triterpenoids. These types of phytochemicals have low polarity and could be dissolved by petroleum ether. Although the phytochemicals contained in the three extraction solution will intersect, however, in general, once a positive result was shown in one extraction solution, the other two extraction solutions do not have to be. The preparation methods of these three extraction solutions are different from those of the following eight solvent extracts (See the supplementary material for details).

Thank you for your explanation of the low content of phenolic compounds in the aqueous extract, so that we could understand the influence of ballast substances on it. But I need to explain that Euphorbia fischeriana is rich in diterpenoids and triterpenoids, there are 176 terpenoids in 250 compounds isolated and identified from Euphorbia fischeriana. And among the fifteen compounds involved in the ten peaks, only two are phenolic compounds, the other 13 compounds were diterpenoids. Therefore, the antioxidant activity of Euphorbia fischeriana mainly comes from terpenes, and the content of total triterpenoids of acetone extract is the highest, which is also an important reason for us to choose acetone extract for further research.

Thank you for pointing out that the water and water-alcohol extracts should be studied more closely, this is also our next plan. Because the traditional application of herbal medicine is water extraction or liquor extraction (be equivalent to water-alcohol extraction). The evaluation of these two extracts can better reflect the traditional efficacy of herbal medicine.

Question 2: Tables 3 and 7 are incorrectly displayed

Answer: Your reminder is very important. We have adjusted the table 3 and table 7 according to your requirements. Now, all the results in table 3 and table 7 could be seen.

Question 3: Experiment planning should be improved and made more consistent. So, for example, the authors chose 8 solvents for the work, but the determination of the classes of compounds was carried out only for 3, not in all tests acetone extracts showed the best results, while LC-MS was carried out only for him. According to the authors, the best solvent is acetone, which is absent in Table 1.

Answer: Dear reviewer, we are sorry that we have not written down the specific experimental methods in our manuscript, which has led to your misunderstanding. Now let us explain our experiment planning. We first conducted qualitative experiments. In qualitative experiment, three kinds of solvents (water, methanol and petroleum ether) were selected to extract medicinal materials, and the extraction solution was directly used to identify fifteen kinds of phytochemicals (See the supplementary material for details),  its experimental results tell us which phytochemicals plants contain and instruct us to carry out the content determination of the components in the next step. In the quantitative experiment and antioxidant experiment, we use eight different solvents to extract the medicinal materials respectively, and then evaporate the solvent under reduced pressure to get the dry extract, and then use these eight dry extracts to carry out the experiment. For example, qualitative experiments show that there are phenolics in the aqueous extract. We then use quantitative experiments to measure the amount of phenolics in each of the eight solvent extracts. Then the antioxidant activity of eight solvent extracts was evaluated, and combined with the results of active ingredient content determination, acetone was finally determined to be the best solvent, acetone extract was worthy of further research.

Round 2

Reviewer 1 Report

The authors have revised the manuscript according to the reviewer's instructions, so I can propose that the manuscript be accepted for publication in Molecules.

Reviewer 2 Report

the authors have worked through all the comments, I think that the work can be published